# Functional and microstructural plasticity following social and interoceptive mental training

Sofie Louise Valk[1,2]*, Philipp Kanske[3,4], Bo-yong Park[5,6,7], Seok-Jun Hong[7,8,9], Anne Böckler[10], Fynn-Mathis Trautwein[11], Boris C Bernhardt[5†], Tania Singer[12†]

[1]Otto Hahn Group Cognitive Neurogenetics, Max Planck Institute for Human Cognitive and Brain Sciences, Leipzig, Germany; [2]INM-7, FZ Jülich, Jülich, Germany; [3]Clinical Psychology and Behavioral Neuroscience, Faculty of Psychology, Technische Universität Dresden, Dresden, Germany; [4]Max Planck Institute for Human Cognitive and Brain Sciences, Leipzig, Germany; [5]Multimodal Imaging and Connectome Analysis Lab, McConnell Brain Imaging Centre, Montreal Neurological Institute and Hospital, McGill University, Montreal, Canada; [6]Department of Data Science, Inha University, Incheon, Republic of Korea; [7]Center for Neuroscience Imaging Research, Institute for Basic Science, Suwon, Republic of Korea; [8]Center for the Developing Brain, Child Mind Institute, New York, United States; [9]Department of Biomedical Engineering, Sungkyunkwan University, Suwon, Republic of Korea; [10]Department of Psychology, Wurzburg University, Wurzburg, Germany; [11]Department of Psychosomatic Medicine and Psychotherapy, Medical Center – University of Freiburg, Faculty of Medicine, University of Freiburg, Freiburg, Germany; [12]Social Neuroscience Lab, Max Planck Society, Berlin, Germany

*For correspondence:
valk@cbs.mpg.de

†These authors contributed equally to this work

Competing interest: The authors declare that no competing interests exist.

**Abstract** The human brain supports social cognitive functions, including Theory of Mind, empathy, and compassion, through its intrinsic hierarchical organization. However, it remains unclear how the learning and refinement of social skills shapes brain function and structure. We studied if different types of social mental training induce changes in cortical function and microstructure, investigating 332 healthy adults (197 women, 20–55 years) with repeated multimodal neuroimaging and behavioral testing. Our neuroimaging approach examined longitudinal changes in cortical functional gradients and myelin-sensitive T1 relaxometry, two complementary measures of cortical hierarchical organization. We observed marked changes in intrinsic cortical function and micro-structure, which varied as a function of social training content. In particular, cortical function and microstructure changed as a result of attention-mindfulness and socio-cognitive training in regions functionally associated with attention and interoception, including insular and parietal cortices. Conversely, socio-affective and socio-cognitive training resulted in differential microstructural changes in regions classically implicated in interoceptive and emotional processing, including insular and orbitofrontal areas, but did not result in functional reorganization. Notably, longitudinal changes in cortical function and microstructure predicted behavioral change in attention, compassion and perspective-taking. Our work demonstrates functional and microstructural plasticity after the training of social-interoceptive functions, and illustrates the bidirectional relationship between brain organisation and human social skills.

## Editor's evaluation

This important work extensively quantifies changes in cortical hierarchical organization induced by different types of social cognitive training. The evidence supporting this is compelling: the authors employ rigorous and complementary multi-modal neuroimaging assessments in a very large sample, measuring longitudinal changes in functional and structural metrics of cortical hierarchical organization. This work has broad applicability to basic neuroscience, illuminating the link between anatomical and functional hierarchies in the brain and social skills, and is also of interest to clinical psychology audiences due to its relevance to interventions such as mindfulness-based therapies.

## Introduction

Humans unique social skills enhance cooperation and survival (*Ochsner and Lieberman, 2001*; *Dunbar, 1998*). Social capacities can be divided into multiple sub-components (*Singer, 2006*; *Schurz et al., 2020*; *Schurz et al., 2021*): (i) socio-affective (or emotional-motivational) abilities such as empathy allowing us to share feelings with others, and may give rise to compassion and prosocial motivation (*Batson, 2009*; *Eisenberg and Fabes, 1990*; *de Vignemont and Singer, 2006*); (ii) socio-cognitive abilities gain access to beliefs and intentions of others [also referred to as Theory of Mind (ToM) or mentalizing (*Singer, 2006*; *Frith and Frith, 2006*; *Saxe and Kanwisher, 2003*)]. Finally, interoceptive abilities, attention, and action observation serve as important auxiliary functions of social aptitudes, contributing to self-other distinction and awareness (*Tomasello, 1995*; *Craig, 2009*; *Kleckner et al., 2017*). These capacities combine externally- and internally-oriented cognitive and affective processes and reflect both focused and ongoing thought processes (*Chun et al., 2011*; *Barrett, 2017*; *Turnbull et al., 2020*; *Murphy et al., 2019*; *Sormaz et al., 2018*). With increasing progress in task-based functional neuroimaging, we start to have an increasingly precise understanding of brain networks associated with the different processes implicated in social cognition. For example, tasks probing socio-emotional functioning and empathy consistently elicit functional activations in anterior insula, supramarginal gyrus, and midcingulate cortex (*Singer, 2006*; *Singer and Lamm, 2009*; *Singer et al., 2004*), while emotional-motivational processes, such as compassion, implicate insular, and orbitofrontal areas (*Lindquist et al., 2012*; *Singer and Klimecki, 2014*). On the other hand, tasks involving socio-cognitive functioning generally activate regions of the human default mode network (DMN), such as medial frontal cortex, temporo-parietal junction, and superior temporal sulcus (*Schurz et al., 2020*; *Saxe and Kanwisher, 2003*; *Bzdok et al., 2012*). Finally, attentional tasks activate inferior parietal and lateral frontal and anterior insular cortices (*Trautwein et al., 2016*; *Corbetta et al., 2008*; *Corbetta and Shulman, 2002*) and interoceptive awareness is linked to anterior insula and cingulate regions (*Craig, 2009*; *Kleckner et al., 2017*; *Seth and Friston, 2016*; *Critchley et al., 2003*). These findings suggest a potentially dissociable neural basis of different social abilities in the human brain.

Despite the progress in the mapping of the functional topography of networks mediating social and interoceptive abilities, the interplay between social behavior and brain organization is less well understood (*Paquola et al., 2022*). Prior research has shown that cortical function and microstructure follow parallel spatial patterns, notably a sensory-transmodal axis that may support the differentiation of sensory and motor function from higher order cognitive processes, such as social cognition (*Valk et al., 2022*; *Park et al., 2021b*; *Paquola et al., 2019b*; *Huntenburg et al., 2017*; *Goulas et al., 2018*). Put differently, a sensory-transmodal framework situates abstract social and interoceptive functions in transmodal anchors, encompassing both heteromodal regions (such as the prefrontal cortex, posterior parietal cortex, lateral temporal cortex, and posterior parahippocampal regions) as well as paralimbic cortices (including orbitofrontal, insular, temporo-polar, cingulate, and parahippocampal regions; *Mesulam, 2000*). Distant from sensory systems, transmodal cortices take on functions that are only loosely constrained by the immediate environment (*Margulies et al., 2016*), allowing internal representations to contribute to more abstract, and social cognition and emotion (*Paquola et al., 2019b*; *Huntenburg et al., 2017*; *Margulies et al., 2016*; *Huntenburg et al., 2018*; *Mesulam, 1998*; *Mesulam, 1994*; *Salehi et al., 2020*; *Cole et al., 2013*; *Beul et al., 2017*; *Barbas, 2015*), thereby enhancing behavioral flexibility (*Mesulam, 1998*; *Murphy et al., 2018*). However, despite the presumed link between cortical microstructure and functional processes it may support, whether and how changes in social behavior impact intrinsic function and microstructure it is not known to date.

Longitudinal investigations can reveal causal links between behavioral skills and brain organization, for example via targeted mental training studies. A range of prior studies indicated that mental training

**eLife digest** Navigating daily life requires a number of social skills, such as empathy and understanding other people's thoughts and feelings. Research has found that specific parts of the brain support these abilities in humans. For instance, the brain areas that support compassion are different from the regions involved in understanding other people's perspective and thoughts.

It is unclear how learning and refining social skills alters the brain. Previous studies have shown that learning new motor skills restructures the areas of the brain that regulate movement. Could acquiring and improving social skills have a similar effect?

To investigate, Valk et al. trained more than 300 healthy adults in different social skills over the course of three months as part of the ReSource project. The program was designed to enhance abilities in compassion and perspective through mental exercises and working in pairs. Participants were also trained using different approaches to see whether changes to the brain are influenced by how a skill is learnt.

The brains of the participants were repeatedly pictured using magnetic resonance imaging (MRI). This revealed that different types of training caused unique changes in specific parts of the brain. For example, teaching mindfulness made parts of the brain less functionally connected, whereas training to understand other people's thought increased functional connections between various regions. These functional alterations were paralleled by changes in brain structure. They could also predict improvements in social skills which were measured throughout the study using behavioural tests.

These findings suggest that training can help to improve social skills even in adults, which may benefit their quality of life through stronger social connections. Better knowledge of how to develop social skills and their biological basis will help to identify people who need support with these interactions and develop new therapies to nurture their abilities.

---

may alter brain function and gross morphology (*Slagter et al., 2007*; *Hölzel et al., 2011*; *Lazar et al., 2005*; *Fox et al., 2016*; *Fox et al., 2014*), but findings do not yet point to a consistent pattern. For example, a randomized controlled trial showed little effect on brain morphology of 8 weeks of mindfulness-based training in healthy adults (*Kral et al., 2022*). Arguably, sample sizes have been relatively modest and training intervals short. Moreover, few studies have compared different practices or focussed on different social skills, despite different types of mental training likely having unique effects on brain and behavior (*Lutz et al., 2021*; *Klimecki et al., 2014*; *Singer and Engert, 2019*). In a previous study realized in the context of the *ReSource* project (*Singer et al., 2016*), our group demonstrated differentiable change in MRI-derived cortical thickness, in support of macrostructural plasticity of the adult brain following the training of social and interoceptive skills (*Valk et al., 2017b*). As the *ReSource* project involved a targeted training of attention-mindfulness (*Presence* training-module, TM), followed by socio-affective (*Affect* TM) and socio-cognitive/ToM training (*Perspective* TM) over the course of nine months, this study design can help to dissociate different mental training effects. Whereas *Presence* aimed at initially stabilizing the mind and nurturing introspective abilities, the *Affect* and *Perspective* TMs focussed on nurturing social skills such as empathy, compassion, and perspective taking on self and others.

Here, we leverage the *ReSource* study dataset to assess whether the targeted training of attention-interoception, socio-affective, and socio-cognitive skills can lead to domain-specific reorganization of (i) intrinsic function (as indexed by resting-state fMRI connectivity gradient analysis), and (ii) cortical microstructure as indexed by quantitative T1 relaxometry, probed along the direction of cortical columns (*Marques et al., 2010*; *Paquola et al., 2020*; *Paquola and Hong, 2023*). Such results would be in line with prior observations suggesting coupled change in brain structure and function (*Mount and Monje, 2017*; *de Faria et al., 2021*), and would help to gain insights into the association between social skills and models of brain organization. Longitudinal analyses of subjects-specific measures of functional integration and segregation evaluated whether these changes corresponded to corresponding change in intracortical microstructure. We also tested for associations to behavioral change in attention, compassion, and ToM markers using machine learning with cross-validation, to evaluate behavioral relevance at the individual level.

## Results

### Embedding of socio-affective and -cognitive functions in cortical brain organization (Fig. 1)

Our work examined changes in brain function and microstructure following social and cognitive mental training. We analyzed resting-state functional MRI (fMRI) measures, myelin-sensitive quantitative T1 (qT1) relaxometry, and behavioral data from 332 adults studied in the ReSource Project (*Singer et al., 2016*). The preregistered trial (https://clinicaltrials.gov/ct2/show/NCT01833104) involved three 3 month long TMs: (i) *Presence*, targeting interoception and attention, (ii) *Affect*, targeting empathy and emotion, and (iii) *Perspective*, targeting ToM. To gain a system-level understanding of brain changes associated with each TM, we took a multi-level approach, combining cortex-wide exploratory analyses of changes in functional and microstructural organization, with an investigation of *a-priori* defined functional networks hypothesized to be targeted by each TM, behavioral prediction of behaviors implicated in each domain.

For *a-priori* functional localization, we selected meta-analytical functional networks mapping these functions using NeuroSynth (*Yarkoni et al., 2011*), (*Figure 1*). To investigate changes in intrinsic functional organization following different types of social and cognitive mental training we focused on changes within a 3D framework of functional axes, explaining in total more than 50% of variance within the functional connectome (*Margulies et al., 2016*; *de Wael et al., 2020*; *Coifman et al., 2005*; *Haak and Beckmann, 2020*; *Bernhardt et al., 2022*). These axes differentiate primary from transmodal cortices (sensory/motor versus abstract cognition, principle gradient, G1), and within this axis further differentiation of visual from sensory-motor regions (secondary gradient, G2), and multiple demand and from default networks (tertiary gradient, G3). To synoptically assess changes within this functional framework, we combined the first three gradients into a marker of functional eccentricity, similar to previous work (*Park et al., 2021a*). Here, regions at either end of the gradient have a high eccentricity, a value based on the average of the three gradients. Following, we investigated gradient-specific effects.

Gradients of each individual were Procrustes aligned to the mean functional connectome based on the human connectome project sample (*Valk et al., 2022*; *Van Essen et al., 2013*), and we calculated region-wise distances to the center of a coordinate system formed by the first three gradients G1, G2, and G3 for each individual [based on the Schaefer 400 parcellation (*Schaefer et al., 2018*)]. Such a gradient eccentricity measures captures intrinsic functional integration (low eccentricity) vs segregation (high eccentricity) in a single scalar value (*Park et al., 2021a*). Highest segregation was observed in visual and sensory-motor networks, while ventral attention and limbic networks were closest to the center of the space. Notably, the *a-priori* networks showed a unique embedding in 3D gradient space (F(5,394) 8.727, $P<0.001$), with *Affect*-associated networks being most integrated while and *Perspective*-networks were most segregated. Studying cortical microstructure, a marker of structural hierarchical organization, we observed patterns of high microstructural integrity (low qT1) in primary areas and low microstructural integrity (high qT1) in transmodal areas, similar to previous work (*Paquola et al., 2019a*; *Burt et al., 2018*). Evaluating *a-priori* network microstructural integrity, we found that compartment 5:12 showed unique microstructural loadings (F(5,394) >5.760, p<0.001), with the emotion meta-analytical network showing lowest microstructural integrity in deep compartments.

### Mental training-specific change in functional organization (Fig. 2)

We first tracked training-related longitudinal changes in functional organization using a holistic and cortex-wide approach through probing the combination of functional gradients 1–3 in functional eccentricity following the different *ReSource* TMs. Following, we investigated specificity of effects in terms of functional gradient and *a-priori* functional networks associated with the TMs. In the *Resource* study, participants were randomly assigned to two training cohorts (TC1, N=80; TC2, N=81), which each underwent a 9-month training consisting of three sequential TMs (*i.e., Presence, Perspective,* and *Affect*) and with weekly group sessions and daily exercises, completed via cellphone and internet platforms (*Figure 1*, *Tables 1–3*, *Materials and Methods* and *Supplementary Materials* for details). TC1 and TC2 underwent the latter two TMs in different order (TC1: *Affect-Perspective*; TC2 *PerspectiveAffect*) to serve as active control groups for each other (*Figure 1A*). Additionally, a matched test-retest control cohort did not undergo any training (RCC, N=90), but

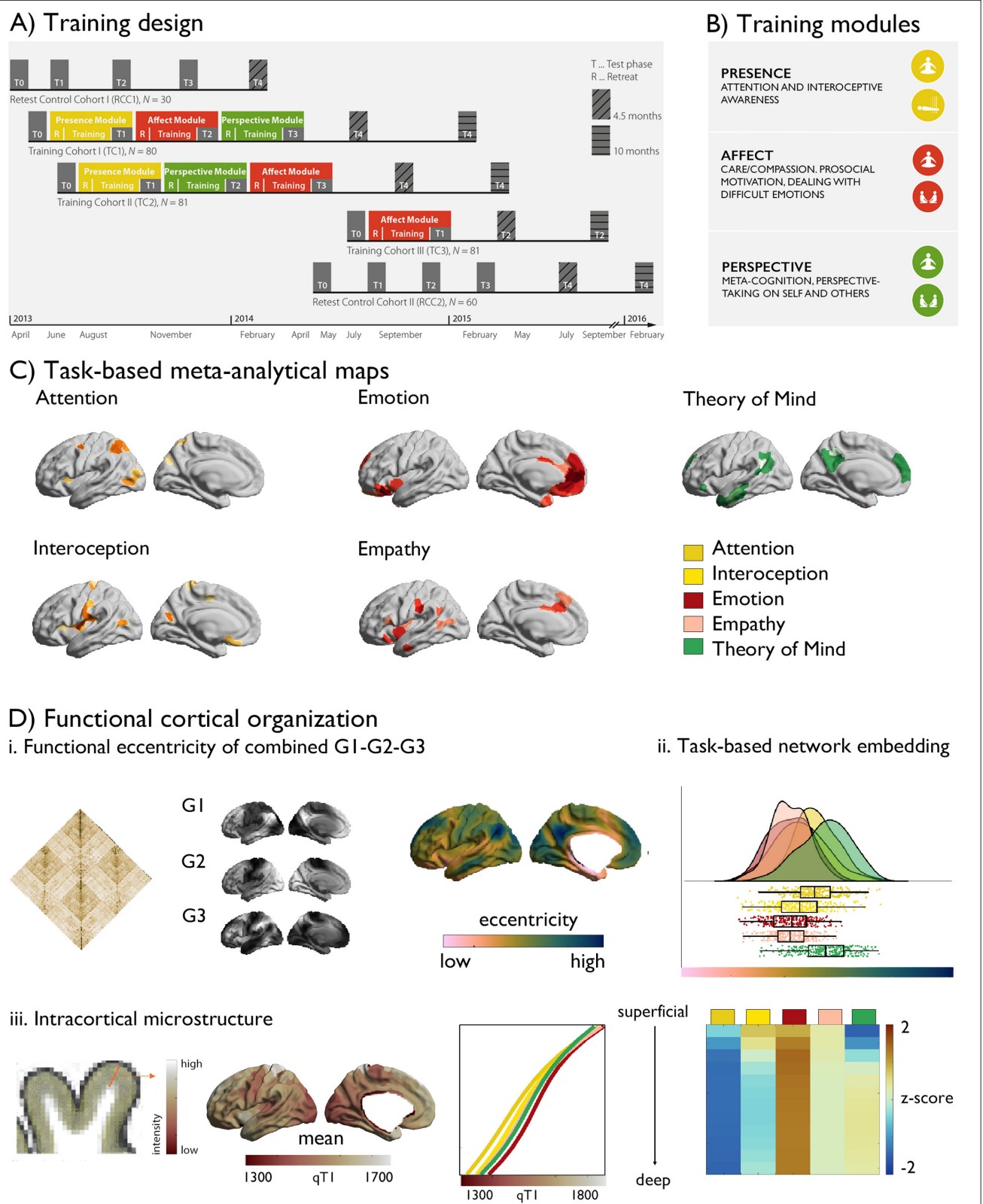

**Figure 1.** Study design. (**A**) Training design of the ReSource study; (**B**) Training modules; (**C**) Task-based meta-analytical maps, and a legend of the color-coding of the maps; (**D**) Functional cortical organization: i. functional connectivity matrix, gradient 1–3, eccentricity metric; ii. task-based network embedding; iii. intracortical microstructure, mean qT1 values as a function of task-based meta-analytical maps and cortical depth and relative values (z-scored per depth-compartment).

**Table 1.** Participant inclusion in resting-state analysis and quantitative T1 analysis.

| Recruited (N, mean age, % female) | $T_0$ (N) | $T_1$ (N) | $T_2$ (N) | $T_3$ (N) |
|---|---|---|---|---|
| Total (N=332) | 268 | 259 | | |
| TC1 (N=80; 41.3; 58.8) | 69 | 65 | 182 | 184 |
| TC2 (N=81; 41.2; 59.3) | 67 | 59 | 57 | 55 |
| RCC (N=90; 40.0; 58.9) | 65 | 68 | 61 | 64 |
| TC3 (N=81; 40.4; 60.5) | 67 | 67 | 64 | 65 |

was followed with the same neuroimaging and behavioral measures as TC1 and TC2. All participants were measured at the end of each three-month TM ($T_1$, $T_2$, $T_3$) using 3T MRI and behavioral measures that were identical to the baseline ($T_0$) measures. There was furthermore an active control group (TC3; N=81), which completed three months of *Affect* training only. In our main analyses, we compared TMs against each other focusing on TMs completed by TC1 and TC2, that is *Presence* ($T_0 \rightarrow T_1$, TC1 and TC2), *Affect* ($T_1 \rightarrow T_2$, TC1 and $T_2 \rightarrow T_3$, TC2), *Perspective* $T_2 \rightarrow T_3$, TC1 and $T_1 \rightarrow T_2$, TC2 and supplementary investigations including also TC3 that only completed a socio-affective training and retest control cohorts.

We evaluated how cortical functional gradients would change following mental training using mixed-effects models (*Dale et al., 1999*). Excluding participants with missing functional or structural data, or excessive movement, the sample included 109 individuals for *Presence*, 104 individuals for *Affect*, 96 individuals for *Perspective*, 168 *retest controls* and 60 *active controls (Affect)* with functional and structural change scores. At the whole-cortex level, we observed marked gradient eccentricity changes following *Presence* and *Perspective* (*Figure 2*, descriptive statistics: *Supplementary file 1a-e*). *Presence* training resulted in increased eccentricity of bilateral temporal and right superior parietal areas (FDRq <0.05), indicative of increased segregation. *Perspective* training resulted in decreased eccentricity of right temporal regions, together with left insular cortices (FDRq <0.05). We observed no eccentricity change following *Affect* training. *Post-hoc* analysis indicated changes between *Presence* and *Perspective* were underlying eccentricity change were most marked in G2 (t=−4.647, p<0.001, d=−0.403), dissociating sensory-motor from visual systems, but not G1 (t=−1.495, p>0.05, d=−0.130) or G3 (t=−0.493, p>0.05, d=−0.043) gradient. Focussing on *a-priori* networks, in particular attention (t=2.842, p=0.005, d=0.247) and interoception (t=2.765, p=0.006, d=0.240) networks showed alterations in *Presence-vs-Perspective*, (*Table 4*, *Figure 2* and *Figure 2—figure supplement 1*). Although effects varied, they were also observed after Global Signal Regression (GSR) control, in TC1 and TC2, and versus RCC (*Supplementary file 1f-j*). Evaluating gradient-specific alterations per *a-priori* network we observed a link between *Presence* versus *Affect* in the empathy-network along G2 (t=3.215, p<0.002; *Supplementary file 1k-m*, *Figure 2—figure supplements 2–4*). Findings were robust when controlling for previously reported cortical thickness change (*Valk et al., 2017b*), *Supplementary file 1n*. We did not find evidence for overall effects of training on functional eccentricity relative to RCC (*Supplementary file 1o*).

**Table 2.** Reason for missing data across the study duration.

*MR incidental findings* are based on $T_0$ radiological evaluations; participants who did not survive *MRI quality control* refers to movement and/or artefacts in the T1-weighted MRI; dropout details can be found in *Singer et al., 2016*; *no MRT*: due to illness / scheduling issues / discomfort in scanner; *other*: non-disclosed; *functional MRI missing:* no complete functional MRI; *functional MRI quality:*>0.3 mm movement (low quality in volume +surface).

| Reason for dropout (TC1, TC2, RCC) | $T_0$ | $T_1$ | $T_2$ | $T_3$ |
|---|---|---|---|---|
| Structural MR incidental finding | 5 | (5 based on $T_0$) | (5 based on $T_0$) | (5 based on $T_0$) |
| Structural MRI quality control | 7 | 6 | 4 | 2 |
| Dropout | 2 | 7 (2 based on $T_0$) | 7 (9 based on $T_{01}$) | 7 (16 based on $T_{012}$) |
| Medical reasons | 1 | 7 (1 based on $T_0$) | 7 (8 based on $T_{01}$) | (15 based on $T_{012}$) |
| Other | 4 | 10 | 7 | 7 |
| Functional MRI missing/QC qT1 | 18 | 14 | 16 | 8 |
| missing | 13 | 7 | 6 | 7 |

**Table 3.** Reason for missing data across the study duration.

*MR incidental findings* are based on $T_0$ radiological evaluations; participants who did not survive *MRI quality control* refers to movement and/or artefacts in the T1-weighted MRI; dropout details can be found in **Singer et al., 2016**; *no MRT*: due to illness / scheduling issues / discomfort in scanner; *other*: non-disclosed.

| Reason for dropout (TC3) | $T_0$ | $T_1$ |
|---|---|---|
| MR incidental finding | 3 | (3 based on $T_0$) |
| MRI quality control | 0 | 0 |
| Dropout | 0 | 3 |
| Medical reasons | 1 | 2 |
| Other | 5 | 3 |
| Functional MRI missing/QC | 1 | 3 |
| qT1 missing | 4 | 0 |

## Overall training effects in microstructure as a function of cortical depth (Fig. 3)

Having established alterations in integration and segregation of *a-priori* networks, we evaluated the neurobiological relevance of these alterations. We investigated changes in cortical microstructure as a function of cortical depth, motivated by the idea that intrinsic functional changes may be anchored in microstructural plasticity that occurs in a depth-varying manner (**Paquola et al., 2022**). Overall, *ReSource* training led to decreased qT1 values, i.e. increased myelination, in both TC1 and TC2 relative to RCC over the nine months training time, in all *a-priori* functional networks in particular in deeper compartment microstructure, whereas RCC showed subtle increases of qT1, suggesting decreased myelination (**Figure 3**, **Supplementary file 1p**, **Figure 3—figure supplement 1**). Studying training-specific effects, we observed marked changes in cortical microstructure following 3-month-long mental training across domains (all FDRq <0.05). *Presence* showed marked increases in qT1 in posterior areas in superficial depth compartments, and marked decreases in qT1 in prefrontal and occipital regions that showed increased spatial extent as a function of cortical depth. Conversely, *Affect* resulted in extended decreases in qT1 in mid and deep depth compartments, in particular in bilateral frontal areas extending to parietal lobe, bilateral posterior cingulate, left fusiform gyrus and right insula. *Perspective* showed largely decreases in qT1 in superficial depths in parietal-temporal, precuneus, and sensory-motor areas, and an increase in qT1 in left prefrontal regions. Patterns were similar when comparing the TMs against each other, highlighting the differentiation between superficial and deep depth-varying changes between *Perspective* and *Affect* and medial prefrontal qT1 decrease following *Presence* relative to *Perspective* and *Affect* as well as *Affect* TC3 and RCC (in particular in case of TC1, **Figure 3—figure supplement 2**).

## Corresponding changes in functional organization and intra-cortical microstructure (Fig. 4)

Having shown alterations in functional and microstructural organization following social mental training, we evaluated corresponding changes in cortical microstructure. A multilevel approach was chosen. First, we evaluated whether the regions observed in functional reorganization in *Presence* versus *Perspective* would also show microstructural change. Second, we studied training-specific microstructural alterations in *a-priori* functional networks. Third, we evaluated the spatial correlation between functional and structural organization as a function of cortical depth. To do so, we sampled qT1 relaxometry values across 12 equidistant intracortical surfaces between the pial and the white matter (**Paquola et al., 2019b**; **Figure 4**). Regions with low mean qT1 were located in sensory-motor and visual regions, regions known to have a high myelin content (**Dinse et al., 2015**; **Sanides and Hoffmann, 1969**). On the other hand, regions with high mean qT1 and thus low myelin content were located in transmodal areas, as previously shown (**Paquola et al., 2020**). We then examined how intra-cortical microstructural organization mirrored observed changes in functional eccentricity in clusters showing differential change during *Presence* vs *Perspective*. We observed a correspondence (FDRq <0.05) between functional eccentricity and upper layer microstructural compartments (1st: t=3.167 p=0.002, d=0.275; 2nd compartment: t=2.911, p=0.004, d=0.253) in regions showing differences in eccentricity between *Presence* and *Perspective*. Following, assessing TM-specific effects in microstructure in *a-priori* task-based functional networks through comparing all TMs, we found all but the emotion network to show increases in qT1 of *Presence* versus *Affect* and *Perspective* in the upper compartments, extending to deeper compartments when comparing *Presence* and

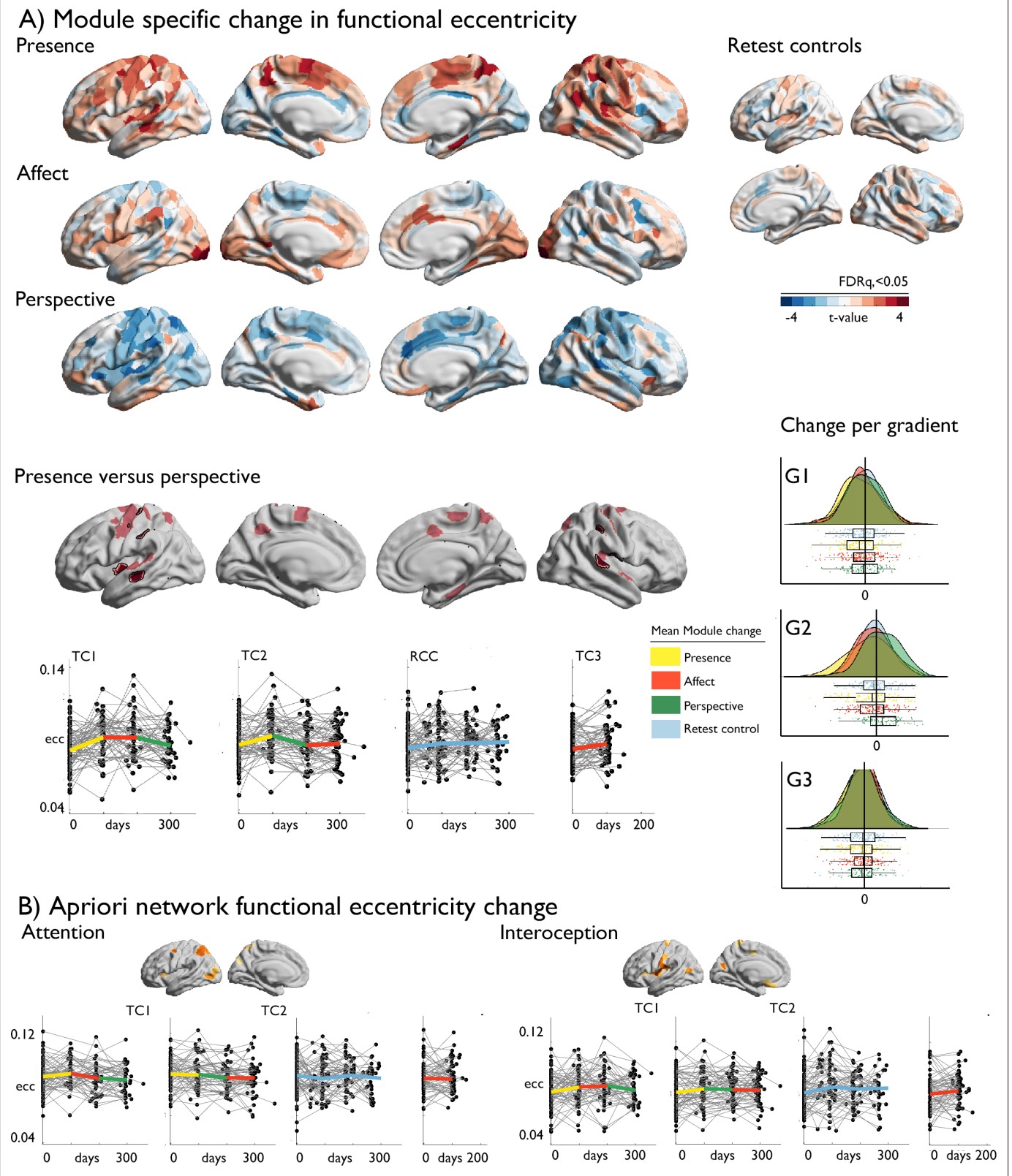

**Figure 2.** Training-induced changes in cortical functional organization. (**A**) *upper:* T-maps of TM-specific changes in functional eccentricity; *lower:* TM-specific change in functional eccentricity, p<0.01, FDRq <0.05 outlined in black, *below:* alterations of eccentricity in the FDRq <0.05 regions, *right:* mean changes in FDRq <0.05 eccentricity regions as a function of G1-G2-G3; (**B**) *A-priori* network functional eccentricity change in networks that showed TM-specific change.

*Figure 2 continued on next page*

*Affect* (FDRq <0.05). Conversely, in deeper compartments, near the GM/WM boundary, we observed decreases of *Affect* relative to *Perspective* in interoception and emotion-related networks (descriptive statistics: ***Supplementary file 1q-u***, ***Figure 4—figure supplement 1***, ***Supplementary file 1v-x***). Findings were largely consistent across the different training cohorts, yet weak relative to retest controls (***Supplementary file 1y-zl***, ***Figure 4—figure supplement 2***). As for the functional change, findings were also observed when controlling for cortical thickness (***Supplementary file 1***: zm-zo), indicating that microstructural change goes above and beyond previously reported morphological change (***Valk et al., 2017b***). Exploring correspondence between functional and microstructural change within TMs, rather than by contrasting TMs, we observed a spatial correlation between functional change in eccentricity and G2 in upper and middle compartment microstructure in *Presence* and overall correspondence with G3 changes, correcting for spatial autocorrelation ($p_{spin}$ <0.05), whereas microstructural alterations in mid- and deeper compartments showed correspondence to eccentricity and G2 in *Affect* ($p_{spin}$ <0.05).

## Functional eccentricity and intracortical microstructure predict social cognitive performance (Fig. 5)

Last, we evaluated whether alterations in cortical microstructure and function following mental training could predict behavioral changes in domains targeted by the TMs. To model changes in brain functional and structural organization, we focused on functional eccentricity, the three gradients, and microstructural depth divided in upper, mid, and deep compartments averaged within *a-priori* functional networks. Previous work has indicated TM specific behavioral changes in attention, compassion, and perspective-taking, as measure using a cued-flanker (attention) and the EmpaTom task (compassion and perspective-taking; ***Trautwein et al., 2020***). Supervised learning (lasso regression, fivefold cross validation, 100 repetitions) with sequential feature selection (7 components, 20% of features) was utilized to predict behavioral change from the average functional gradient eccentricity, and G1-G3, as well as microstructure in superficial (1:4), middle (5:8), and deep (9:12) compartments in the five *a-priori* networks, resulting in 35 features to select from (***Figure 5***). We incorporated age and sex regression into the cross-validation model, to avoid leakage. Attention change predictions in *Presence* (N=85, TC1 and TC2) were most marked in attention network in microstructure at superficial depts and eccentricity (nMAE (mean ± SD): –0.037±0.003, out of sample r (mean ± SD): 0.325±0.308). Conversely, compassion change following *Affect* (N=100, TC1 and TC2) was predicted primarily through structural and functional reorganization of attention, interoception and emotion networks (MAE: –0.412±0.029, out of sample r: 0.284±0.279). Last, Theory of Mind change following *Perspective* (N=93, TC1 and TC2) predictions (nMAE: –0.081±0.006 out-of-sample r: 0.301±0.281) were most

**Table 4.** TM-specific changes in eccentricity per a priori networks.

|  | Presence (n=109) vs Perspective (n=96) | Presence (n=109) vs Affect (n=104) | Perspective (n=96) vs Affect (n=104) |
|---|---|---|---|
| Attention | t=2.842, p=0.005*, d=0.247 | t=1.458, p>0.05, d=0.127 | t=−1.692, p>0.05, d=−0.147 |
| Interoception | t=2.765, p=0.006*, d=0.240 | t=1.043, p>0.05, d=0.091 | t=−2.008, p=0.045, d=−0.174 |
| Emotion | t=0.387, p>0.05, d=0.035 | t=−0.135, p>0.05, d=−0.011 | t=−0.552, p>0.05, d=−0.048 |
| Empathy | t=2.218, p=0.027, d=193 | t=0.879, p>0.05, d=0.076 | t=−1.569, p>0.05, d=−0.136 |
| Theory of Mind | t=1.721, p>0.05, d=0.149 | t=1.324, p>0.05, d=0.115 | t=−0.601, p>0.05, d=−0.052 |

*signifies FDR corrected differences.

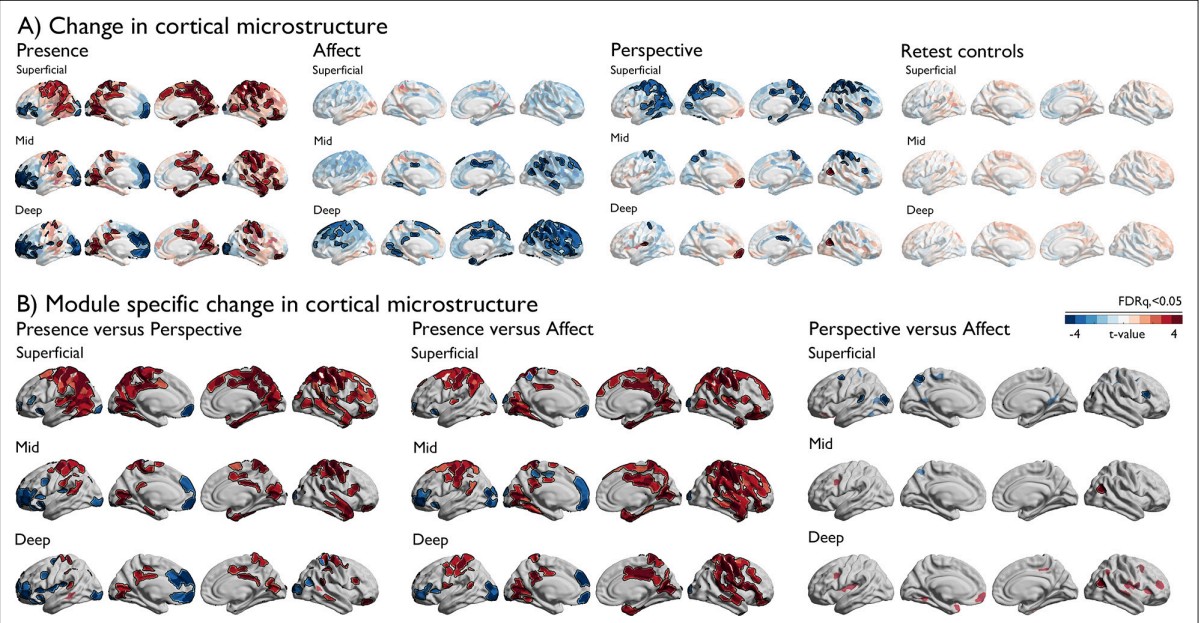

**Figure 3.** Changes in depth-varying microstructure as a function of TM. (**A**). Change in cortical microstructure, per TM, red indicates positive change in qT1, blue negative change. FDRq <0.05 findings are outlined in black on top of t-values per parcel; (**B**) TM specific change in cortical microstructure. Red indicates positive change in qT1, blue negative change. FDRq <0.05 findings are outlined in black in combination with semi-transparent trends (p<0.01).

The online version of this article includes the following figure supplement(s) for figure 3:

**Figure supplement 1.** Training-specific change in qT1 from baseline to T3 as a function superficial (1:4) mid (5:8) and deep (8:12) depth compartment.

**Figure supplement 2.** Cohort-specific change in qT1 from baseline to $T_1$ as a function superficial (1:4) mid (5:8) and deep (8:12) depth compartment.

likely to occur in attention networks along G3 and microstructure of upper and middle compartments, as well as emotion related networks along G3. To further test our predictive models, we evaluated model performance on random test data, as well as non-domain behavioral scores. We found that in all cases, the domain-specific TM model performed best on test data of the respective TM (p<0.001).

## Discussion

We studied whether targeted training of human (social)cognitive and affective skills would alter intrinsic functional and structural organizational axes in a systematic and domain-specific manner. We evaluated longitudinal changes in MRI-derived cortical functional gradients and qT1 profiles as well as their interrelationship in the context of the 9-month *ReSource* study (*Singer et al., 2016*). We demonstrated intrinsic functional and microstructural plasticity that varied as a function of distinct social mental trainings, and that can predict training-related behavioral change. In particular, training attention/mindfulness, emotion/motivation, and socio-cognitive skills led to differential changes in *a-priori* network integration/segregation anchored in the secondary gradient differentiating sensory modalities. Moreover, functional changes showed correspondence to microstructural changes as a function of cortical depth and content of training. In sum, here we provide longitudinal evidence of a relationship between human social behaviors and intrinsic cortical function and depth-varying microstructure.

Analyzing meta-analytical fMRI networks involved in social and attentional processes at the baseline time point (i.e. before the ReSource training started), we demonstrated that networks associated with social processing were differentially positioned in a coordinate system spanned by the first three functional gradients. Echoing a mounting literature (*Margulies et al., 2016*; *Bernhardt et al., 2022*; *Hong et al., 2019*; *Bijsterbosch et al., 2021*), the principal gradient ran from unimodal to transmodal systems with as apex the DMN. This axis aligns with classic notions of the primate cortical hierarchy (*Mesulam, 1998*; *Markov and Kennedy, 2013*), axes of microstructural differentiation

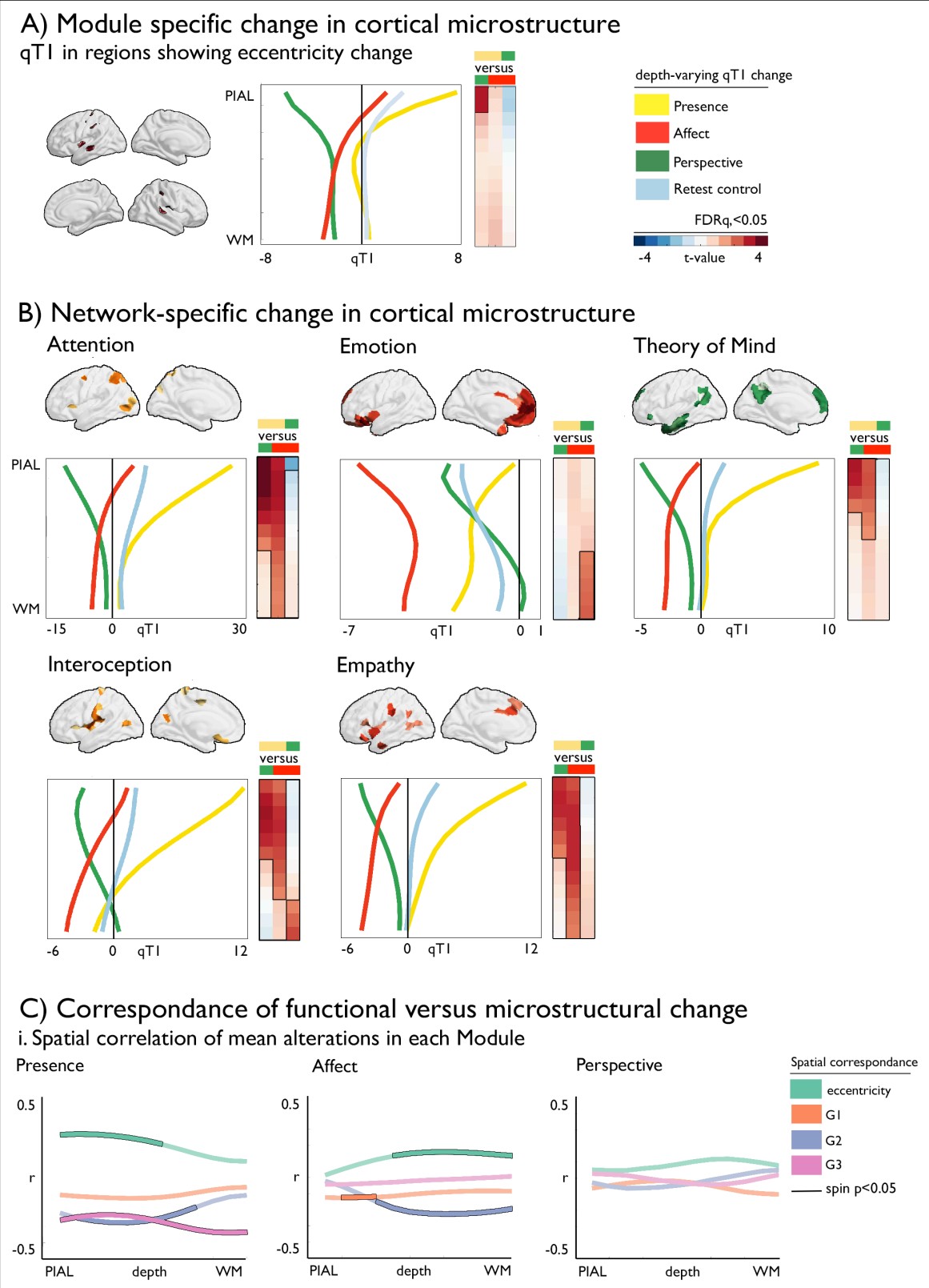

**Figure 4.** Dissociable microstructural alterations following mental training. (**A**).TM-specific changes in cortical microstructure; qT1 in regions showing eccentricity change (y-axis: depth, x-axis: qT1 change); (**B**) Network-specific change in cortical microstructure as a function of depth, mean change per TM, p_FDR <0.05 have black outline (y-axis: depth, x-axis: qT1 change). The boxes on the right of each plot display the statistics (t-values) of the

*Figure 4 continued on next page*

*Figure 4 continued*

respective difference between TM, with the contrast color coded as upper minus lower TM (defined by color); (**C**) Correspondence of functional versus microstructural change; i. Spatial correlation of mean alterations in each TM, black outline indicates p$_{spin}$ <0.05, as a function of cortical depth.

The online version of this article includes the following figure supplement(s) for figure 4:

**Figure supplement 1.** TM-specific change in superficial (1:4) mid (5:8) and deep (8:12) depth compartment microstructure as a function of training cohort and *a-priori* network.

**Figure supplement 2.** TM-specific change in qT1 as a function superficial (1:4) mid (5:8) and deep (8:12) depth compartment.

(*Paquola et al., 2019b*; *Huntenburg et al., 2017*; *Paquola et al., 2020*) and cortical evolution, with heteromodal regions undergoing recent expansions in the human lineage (*Goulas et al., 2018*; *Xu et al., 2020*; *Valk et al., 2020*; *Changeux et al., 2021*). Conversely, the second gradient dissociates visual and sensory systems and the tertiary gradient dissociates the task-positive, attention and control, networks from rest of the brain (*Margulies et al., 2016*). Unlike the DMN, the task-positive network, including frontal and parietal regions, engages preferentially in externally-oriented tasks (*Buckner et al., 2008*; *Fox et al., 2005*; *Duncan, 2010*). This axis may differentiate but between DMN-related socio-episodic memory processing from task-focused processing associated with the multiple demand network (*Turnbull et al., 2020*; *Turnbull et al., 2019*). Together, the three gradients describe a processing organization, with primary systems and DMN regions showing functional segregation and saliency network functional integration (*Park et al., 2021a*; *Bethlehem et al., 2020*; *Smallwood et al., 2021*). Indeed, along its axes, we found *a-priori* networks associated with ToM and

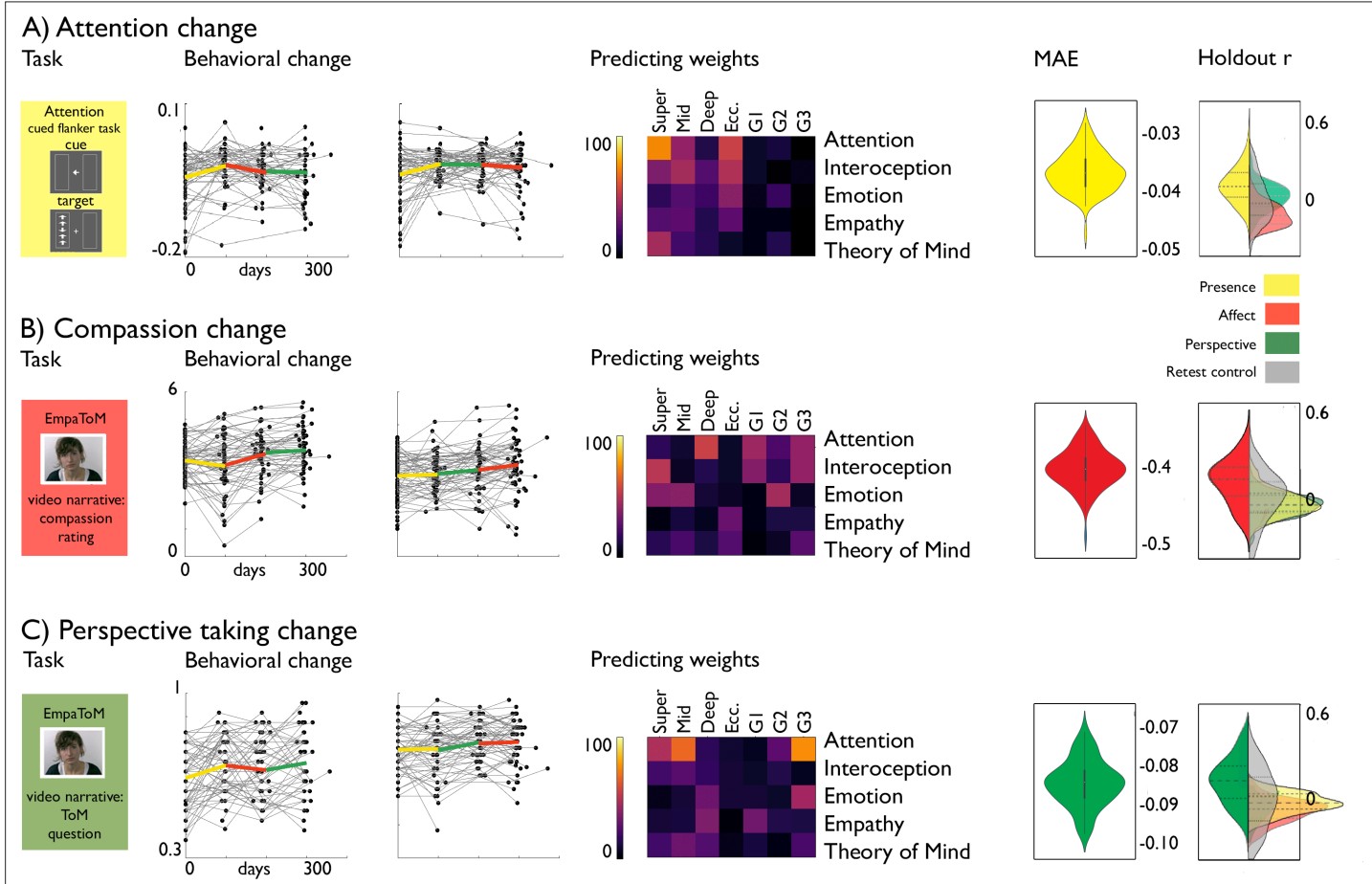

**Figure 5.** Behavioral change prediction. (**A**).Attention change; from *left* to *right*: attention task, behavioral change, predicting weights, nMAE and holdout r distribution; (**B**). Compassion change; from *left* to *right*: compassion task, behavioral change, predicting weights, nMAE and holdout r distribution; (**C**). Perspective-taking change; from *left* to *right*: perspective-taking task, behavioral change, predicting weights, nMAE and holdout r distribution (left side; within TM, right side; other TM (yellow: *Presence*; red: *Affect*; Green: *Perspective*) or RCC (grey)).

attention to be most segregated, indicating that their functional architecture is distinct from other transmodal networks. Conversely, emotion-related, and to a, lesser extent empathy and interoceptive processing, *a-priori* networks were found to be relatively integrated with all other networks. This positioning may be reflective of a regulative role of functional activity of task-positive and -negative networks (*Menon and Uddin, 2010*). As such, we could show that the meta-analytical functional activations associated with social cognition each were placed at unique locations along cardinal axes of functional organization, showing varying levels of functional integration and segregation.

Studying functional network plasticity following ReSource training, we found that regions associated with attention and interoception showed functional integration following Perspective training, whereas they segregated following *Presence*. In the context of social cognition, integrated states might reflect active thought processes such as those required when engaging in ToM, while more segregated states may reflect automation and domain-specific function (*Finc et al., 2020*). Indeed, attention-based mindfulness, as cultivated during *Presence*, may reduce habitual thought patterns and enhance momentary awareness (*Lutz et al., 2019*) – possibly captured by functional network segregation. Moreover, our observations in the domain of social skill training align well with a previous observation that working memory task performance is associated with decreased modularity and increased integration, whereas automation following working memory training was associated with increased modularity and segregation of multiple demand network and DMN (*Finc et al., 2020*). Second, 8-week simultaneous training of compassion, mindfulness and perspective taking has been reported to result in reduced intra-network connectivity in the DMN, VAN, and somatomotor networks, reflecting integration (*Cotier et al., 2017*). Interestingly, overall changes in functional integration and segregation seemed to be most strongly related to loadings of the secondary gradient. This could imply a putative role of sensory-motor integration or shift in sensory-association patterning between sensory modalities as a function of mental training (*Kerr et al., 2013*). Moreover, the secondary gradient has been implicated in guiding task-activation changes linked on- and off-task thought over time (*Turnbull et al., 2020*). More generally, our findings may be in line with the Global Workspace Theory of cognition, which poses that automated tasks, such as interoception and awareness, can be performed within segregated clusters of regions, whereas those that are challenging, for example perspective-taking, require integration (*Dehaene et al., 1998*). Of note, we did not observe changes in eccentricity following the *Affect* TM, relative to the other TMs. Rather, *Affect* seemed to stabilize changes in eccentricity observed following the other trainings. It is possible that the lack of change in eccentricity following socio-affective training reflects a coordinating role of socio-affect relative to alterations associated with attention-mindfulness and socio-cognition. Such an interpretation aligns with theories of emotional allostasis, suggesting that affective processing may balance integration and segregation of brain function to regulate resources dynamically (*Chun et al., 2011*; *Barrett, 2017*). Notably, though the current work focused on cortical networks, subcortical regions are known to be core constituents of the functional organization of the social brain (*Shine, 2021*; *Kanske et al., 2015*; *Adolphs, 2009*). Follow-up work that studies plasticity of subcortical function and structure in the context of the social brain may provide additional system-level insights.

To further evaluate potential neurobiological substrates of cortical functional reorganization, we leveraged intra-cortical microstructure profiling using equidistant probes perpendicular to the cortical mantle (*Paquola et al., 2019a*; *Waehnert et al., 2014*). This targeted changes in cortical myeloarchitecture, to expand upon recent reports of experience-dependent myelin plasticity (*Mount and Monje, 2017*; *de Faria et al., 2021*) as well as our previous finding of macrostructural change in cortical morphology the same cohort (*Valk et al., 2017b*). Evaluating functional eccentricity alterations in attention and interoception networks observed between *Presence* and *Perspective,* we found that these changes could be particularly recapitulated by microstructural alterations in superficial cortical depths. Whereas socio-cognitive training resulted in increased myelination proxy (reflected by decreases in qT1) values, attention-based mindfulness training resulted in decreased myelination of these regions. Similar patterns were observed in empathy and Theory of Mind networks. Contrasting attention-mindfulness and social affective training, we observed similar patterns of changes in microstructure in superficial compartments, which extended to middle compartments for attention, interoception and empathy-related meta-analytical *a-priori* functional networks. Conversely, we found differential alterations of qT1 comparing socio-cognitive with affective training in deep compartments of interoceptive and emotion-related functional networks, where socio-affective training resulted in

relative increase in myelin as compared to socio-cognitive training. Various studies have reported training-induced changes in microstructure in humans, often measured using diffusion MRI derivative measures, such as fractional anisotropy (*Taubert et al., 2010*; *Takeuchi et al., 2010*; *Scholz et al., 2009*). Work in mice has shown that social isolation during development resulted in changes in mPFC myelination, correlating with working memory and social behaviors (*Makinodan et al., 2012*). The observed alterations in qT1 in our study may reflect changes in oligodendrites as well as supporting glial cells (*Bonetto et al., 2020*; *Chorghay et al., 2018*). Indeed, myelin-plasticity has been suggested to be a key biological marker of learning and memory in adulthood (*de Faria et al., 2021*; *Xin and Chan, 2020*). Due to an overall lower myelination of association regions, conduction velocities may still be adjusted dynamically in adulthood, as a function of changing environmental demands (*Mount and Monje, 2017*). Interestingly, alterations persisted above and beyond cortical thickness change alone (*Valk et al., 2017b*). As structural plasticity includes both synaptic and myelin-related processes, further study on the plasticity of multiple microstructural markers in vivo may help to disentangle the different biological processes that underlie adaptive cognition in adulthood.

Co-alterations of function and structure following different types of mental TMs were further evidenced by TM specific analyses, including behavioral predictions. Together, they suggest co-alterations of function and microstructure to occur in a manifold fashion, with differential change occurring both in spatially divergent functional networks and as a function of different functional axes and cortical depts. By integrating functional organization with intra-cortical microstructure in the context of a longitudinal study involving social and cognitive mental training in adults, the current study integrates perspectives on structure and functional organization of the human cortex and its relation to cognition. The layered structure of the human cortex is a key element supporting its function, leading up to different functional organizational dimensions (*Turnbull et al., 2020*; *Goulas et al., 2018*; *Turnbull et al., 2019*). In particular, the layered structure of the cortex enables feedforward and feedback information transfer between regions in a hierarchical manner (*Felleman and Van Essen, 1991*). Whereas ascending connections have been associated with middle cortical layers (origin: layer 3, termination: layer 4), descending connectivity links to superficial and deeper layers (origin: layer 2, 3 A and 6, termination: layer 1 and 6) (*Bastos et al., 2012*; *Rockland, 2019*; *Rockland and Pandya, 1979*). Importantly, there may be key differences between layer-wise connectivity in primary and transmodal areas (*Rockland, 2019*; *Finn et al., 2021*). Functionally, it has been suggested that, in humans, superficial layers of the transmodal cortex have been implicated in manipulating information, whereas deeper layers may have a more controlling function (*Bastos et al., 2018*; *Finn et al., 2019*). Interestingly, using high-resolution data it has been shown that frontal regions dominate feedback processes, whereas parietal regions support feedforward functions (*Huber et al., 2021*). Such differences may relate to the differential pattern in anterior regions associated with emotion processing, showing somewhat distinct patterning relative to posterior regions across mental-training content. Moreover, various studies in non-human mammals have indicated differential mechanisms of plasticity correspond to different layer-depths (*Verhoog et al., 2016*; *Goel and Lee, 2007*; *Yang et al., 2004*), underscoring the relevance of depth-dependent variation in cortical microstructure over time. It is of note that the current study used a proxy of layer-depth that does not correspond to actual cortical layers, but rather creates depth-varying profiles of 1 mm resolution quantitative qT1 though interpolation. However, depth-variations in such profiles have been used to quantify individual differences in both histological sections (*Amunts et al., 1999*) and in vivo during development (*Paquola et al., 2019a*). Our work highlighted depth-dependent changes in cortical microstructure, with associations to intrinsic functional and behavioral change that go above and beyond cortical thickness alterations. Yet, we cannot exclude that part of the effects also include partial volume averaging, given the resolution of the data and study set-up. Further work will benefit from including longitudinal paradigms with sub-millimeter microstructural and intrinsic functional markers to further understand the interplay between depth-varying microstructure, its plasticity, and intrinsic brain function.

In sum, combining a longitudinal mental training study with multi-modal imaging, we could show that mental TMs focusing on attention, socio-emotional and socio-cognitive skills resulted in differentiable change in intrinsic functional and microstructural organization. In line with prior work revealing differential changes in grey matter morphology after each of the three *ReSource* TMs in the same sample (*Valk et al., 2017b*), the current work differentiates processes related to our ability of understanding the thoughts and feelings of ourselves and others within the intrinsic functional and

microstructural organization of the human brain, as such, our findings are compatible with previous work suggesting a link between anatomical and functional hierarchies and global workspace theories of cognition (*Dehaene et al., 1998*; *Baars, 2002*; *Deco et al., 2021*). Although our work focused on healthy adults ranging from 20 to 55 years of age, our findings overall support the possibility that targeted mental training can enhance social skills and lead to co-occurring reconfigurations of cortical function and microstructure, providing evidence for experience-dependent plasticity.

## Materials and methods
### Experimental design
The specifics on the experimental design are the similar to related works in the same sample (*Valk et al., 2017b*; *Trautwein et al., 2020*). They are provided again here for completeness.

### Participants
A total of 332 healthy adults (197 women, mean ± SD = 40.7±9.2 years, 20–55 years), recruited in 2012–2014 participated in the study, see *Table 1* for more details. More than 95% of our sample was Caucasian, with catchment areas balanced across two German municipalities (Berlin and Leipzig). Participant eligibility was determined through a multi-stage procedure that involved several screening and mental health questionnaires, together with a phone interview (for details, see *Singer et al., 2016*). Next, a face-to-face mental health diagnostic interview with a trained clinical psychologist was scheduled. The interview included a computer-assisted German version of the Structured Clinical Interview for DSM-IV Axis-I disorders, SCID-I DIA-X (*Wittchen and Pfister, 1997*) and a personal interview, SCID-II, for Axis-II disorders (*Weissman et al., 1997*; *First et al., 1997*). Participants were excluded if they fulfilled criteria for: (i) an Axis-I disorder within the past two years; (ii) Schizophrenia, psychotic disorders, bipolar disorder, substance dependency, or an Axis-II disorder at any time in their life. No participant had a history of neurological disorders or head trauma, based on an in-house self-report questionnaire used to screen all volunteers prior to imaging investigations. In addition, participants underwent a diagnostic radiological evaluation to rule out the presence of mass lesions (e.g., tumors, vascular malformations). All participants gave written informed consent and the study was approved by the Research Ethics Committees of the University of Leipzig (#376/12-ff) and Humboldt University in Berlin (#2013–02, 2013–29, 2014–10). The study was registered at ClinicalTrials.gov under the title 'Plasticity of the Compassionate Brain' (#NCT01833104). For details on recruitment and sample selection, see the full cohort and study descriptor (*Singer et al., 2016*).

### Sample size estimation and group allocation
Overall, 2595 people signed up for the *ReSource* study in winter 2012/2013. Of these individuals, 311 potential participants met all eligibility criteria. From the latter group, 198 were randomly selected as the final sample. Participants were selected from the larger pool of potential participants and assigned to cohorts using bootstrapping without replacement, creating cohorts that did not differ (omnibus test *P*<0.1) in demographics (age, gender, marital status, income, and IQ) or self-reported traits (depression, empathy, interoceptive awareness, stress level, compassion for self and others, alexithymia, general mental health, anxiety, agreeableness, conscientiousness, extraversion, neuroticism, and openness). Seven participants dropped out of the study after assignment but before data collection began, leaving 30 participants in RCC1, 80 in TC1, and 81 in TC2.

A total of 2144 people applied for the second wave of the study in winter 2013/2014. Of these people, 248 potential participants met all the eligibility criteria. From the latter pool, 164 were then randomly selected as the final sample. Participants were selected from the larger pool of potential participants and assigned to cohorts using bootstrapping without replacement, creating cohorts that did not differ significantly (omnibus test, p>0.1) from the Winter 2012/2013 cohorts or from one another in demographics (age, gender, marital status, income, and IQ) or self-reported traits (depression, empathy, interoceptive awareness, stress level, compassion for self and others, alexithymia, general mental health, anxiety, agreeableness, conscientiousness, neuroticism, and openness). The control cohorts (RCC1, RCC2, and RCC1&2) were significantly lower in extraversion than TC3; participants in the control cohorts were also more likely to have children than participants in

TC3. Twenty-three participants dropped out of the study after assignment but before data collection began, leaving 81 participants in TC3 and 60 in RCC2. See further (*Singer et al., 2016*).

## Study design

Our study focused on two training groups: training cohort 1 (TC1, n=80 at enrolment) and training cohort 2 (TC2, n=81 at enrolment), as well as a retest control cohort (RCC) that was partly measured prior to (n=30 at enrolment) and partly after (n=60 at enrolment) TC1 and TC2. A third training cohort (TC3, n=81 at enrolment) underwent an independent training program, and was included as an additional active control for the *Presence* TM. Participants were selected from a larger pool of potential volunteers by bootstrapping without replacement, creating cohorts not differing significantly with respect to several demographic and self-report traits (*Singer et al., 2016*). Total training duration of TC1 and TC2 was 39 weeks (~nine months), divided into three TMs: *Presence*, *Affect*, and *Perspective* (see below, for details), each lasting for 13 weeks (*Figure 1*). TC3 only participated in one 13 week *Affect* training, and is only included in supplementary robustness analyses, so that the main analysis of functional plasticity focusses on TC1 and TC2. Our main cohorts of interest, TC1 and TC2, underwent *Affect* and *Perspective* TMs in different order to act as active control cohorts for each other. Specifically, TC1 underwent '*Presence-Affect-Perspective*', whereas TC2 underwent '*Presence-Perspective-Affect*'. TC1, TC2, and RCC underwent four testing phases. The baseline-testing phase is called T0; testing phases at the end of the xth TM are called Tx (*i.e.*, T1, T2, T3). In RCC, testing was carried out at similarly spaced intervals. The study had a slightly time-shifted design, where different groups started at different time points to simultaneously accommodate scanner and teacher availability. As we focused on training-related effects, we did not include analysis of a follow-up measurement T4 that was carried out 4, 5, or 10 months after the official training had ended. For details on training and practice set-up, timeline, and measures, see *Singer et al., 2016*.

## Final sample

We excluded individuals with missing structural and/or functional MRI data and/or a framewise-displacement of >0.3 mm (<5%) (*Power et al., 2012*). Further details of sample size per time-point and exclusion criteria are in *Tables 1–3*.

## Neuroimaging acquisition and analysis

### MRI acquisition

MRI data were acquired on a 3T Siemens Magnetom Verio (Siemens Healthcare, Erlangen, Germany) using a 32-channel head coil. We recorded task-free functional MRI using a T2*-weighted gradient 2D-EPI sequence (repetition time [TR]=2000ms, echo time [TE]=27ms, flip angle = 90°; 37 slices tilted at approximately 30° with 3 mm slice thickness, field of view [FOV]=210 × 210mm2, matrix = 70 × 70, 3×3 × 3 mm3 voxels, 1 mm gap; 210 volumes per session). We also acquired a T1-weighted 3D-MPRAGE sequence (176 sagittal slices, TR = 2300ms, TE = 2.98ms, inversion time [TI]=900ms, flip angle = 7°, FOV = 240 × 256 mm$^2$, matrix = 240 × 256, 1×1 × 1 mm$^3$ voxels). Throughout the duration of our longitudinal study, imaging hardware and console software (Syngo B17) were held constant. For qT1 mapping, we used the recently introduced 3D MP2RAGE sequence (*Marques et al., 2010*), which combines two MPRAGE datasets acquired at different inversion times to provide intrinsic bias field cancellation and estimation of T1 TR = 5000ms, TE = 2.89ms, TI = 700/2500ms, flip angle = 4/5°, FOV = 256 × 256 mm2, 1×1 × 1–mm3 voxels.

### Task-free functional MRI analysis

Processing was based on DPARSF/REST for Matlab [http://www.restfmri.net (*Song et al., 2011*)]. We discarded the first 5 volumes to ensure steady-state magnetization, performed slice-time correction, motion correction and realignment, and co-registered functional time series of a given subject to the corresponding T1-weighted MRI. Images underwent unified segmentation and registration to MNI152, followed by nuisance covariate regression to remove effects of average WM and CSF signal, as well as 6 motion parameters (3 translations, 3 rotations). We included a scrubbing (*Power et al., 2012*) that modelled time points with a frame-wise displacement of ≥0.5 mm, together with the

preceding and subsequent time points as separate regressors during nuisance covariate correction. Volume-based timeseries were mapped to the fsaverage5 surface using bbregister.

## Gradient construction

In line with previous studies evaluating functional gradients (*Paquola et al., 2019b*; *Margulies et al., 2016*; *de Wael et al., 2020*; *Hong et al., 2019*; *Bethlehem et al., 2020*; *Vos de Wael et al., 2018*) the functional connectivity matrix was proportionally thresholded at 90% per row and converted into a normalised angle matrix using the BrainSpace toolbox for MATLAB (*de Wael et al., 2020*). Diffusion map embedding, a nonlinear manifold learning technique (*Coifman et al., 2005*), identified principal gradient components, explaining functional connectivity variance in descending order. In brief, the algorithm estimates a low-dimensional embedding from a high-dimensional affinity matrix. In this space, cortical regions that are strongly interconnected by either many suprathreshold edges or few very strong edges are closer together, whereas nodes with little or no functional connectivity are farther apart. The name of this approach, which belongs to the family of graph Laplacians, derives from the equivalence of the Euclidean distance between points in the embedded space and the diffusion distance between probability distributions centred at those points. It is controlled by the parameter α, which controls the influence of the density of sampling points on the manifold ($\alpha=0$, maximal influence; $\alpha=1$, no influence). Based on previous work (*Margulies et al., 2016*), α was set at 0.5, this retains the global relations between data points in the embedded space and has been suggested to be relatively robust to noise in the functional connectivity matrix. The diffusion time (t), which controls the scale of eigenvalues of the diffusion operator was set at t=0 (default). Individual embedding solutions were aligned to the group-level embedding based on the Human Connectome Project S1200 sample (*Van Essen et al., 2013*) via Procrustes rotations (*de Wael et al., 2020*). The Procrustes alignment enables comparison across individual embedding solutions, provided the original data is equivalent enough to produce comparable Euclidean spaces.

## 3D gradient metric: eccentricity

To construct the combined gradient, we computed the Euclidean distance to the individual center for gradient 1–3. Next, to evaluate change within and between individuals, we computed the difference between gradient scores between different time-points.

## Processing of microstructural data

The details on the processing are identical to previous work (*Paquola et al., 2019b*; *Valk et al., 2022*). We have provided the information here again for clarity. T1-weighted MRIs were processed using FreeSurfer (http://surfer.nmr.mgh.harvard.edu) version 5.1.0 to generate cortical surface models for measurements of cortical thickness and surface area. FreeSurfer has been validated against histological analysis (*Rosas et al., 2002*) and manual measurements (*Kuperberg et al., 2003*). We chose the most general cross-sectional image processing procedure to enable the longitudinal as well as cross-sectional study goals of the ReSource Project see for example (*Valk et al., 2017a*). Since data acquisition spanned more than 2 years, a non-specific imaging procedure enabled baseline data analysis before the completion of latter time points, without compromising comparability. Quantitative T1 maps were aligned and mapped to the T1-weighted MRI using a boundary-based registration (*Greve and Fischl, 2009*), by maximizing the intensity gradient across tissues boundaries and using the surfaces that separate brain structure and tissue types of the T1-weighted reference image, and the tissue intensity of the quantitative T1 map. Following we computed a dept-dependent microstructure proxy. Depth-dependent cortical microstructure analysis has a long tradition in neuroanatomy (*Schleicher et al., 1999*; *Zilles et al., 2002*), and depth-dependent shift in cellular and myelin characteristics have been shown to reflect architectural complexity (*Zilles et al., 2002*) and cortical hierarchy (*Mesulam, 1998*). We generated 12 equivolumetric surfaces between outer and inner cortical surfaces. The equivolumetric model adjusts for cortical folding by varying the Euclidean distance $\rho$ between pairs of intracortical surfaces throughout the cortex to preserve the fractional volume between surfaces. $\rho$ was calculated for each surface (*Ochsner and Lieberman, 2001*):

$$\rho = \frac{1}{A_{out} - A_{in}} \cdot \left( -A_{in} + \sqrt{\alpha A_{out}^2 + (1 - \alpha) A_{in}^2} \right)$$

in which α represents a fraction of the total volume of the segment accounted for by the surface, while $A_{out}$ and $A_{in}$ represents the surface area of outer and inner cortical surfaces, respectively. We systematically sampled qT1 values for each of the 20,484 linked vertices from the outer to the inner surface across the cortex. Following, we computed the average value of qT1 in each of the 400 parcels (*Schaefer et al., 2018*).

## Meta-analytical a-priori networks

We used the NeuroSynth meta-analytic database (http://www.neurosynth.org) (*Yarkoni et al., 2011*) to assess topic terms associated with the training ('attention', 'interoception', 'emotion', 'empathy', 'Theory of Mind').

## Behavioral markers

We assessed a battery of behavioral markers developed and adapted to target the main aims of the *Presence, Perspective*, and *Affect* TMs: selective attention, compassion, and ToM. Behavioral changes of these markers elicited by the different TMs are reported elsewhere (*Trautwein et al., 2020*). The measure for compassion was based on the EmpaToM task, a developed and validated naturalistic video paradigm in the current subjects (*Kanske et al., 2015*; *Tholen et al., 2020*). Videos showed people recounting autobiographical episodes that were either emotionally negative (e.g. loss of a loved one) or neutral (e.g. commuting to work), followed by Likert-scale ratings of experienced valence and compassion. Since the conceptual understanding of compassion might change due to the training, we ensured a consistent understanding by defining it prior to each measurement as experiencing feelings of care, warmth, and benevolence. Compassion was quantified as mean of compassion ratings across all experimental conditions. The EmpaToM task (*Kanske et al., 2015*) also allowed for measurement of ToM performance. After the ratings, multiple-choice questions requiring inference of mental states (thoughts, intentions, beliefs) of the person in the video or factual reasoning on the video's content (control condition) were asked. Questions had three response options and only one correct answer, which had been validated during pre-study piloting (*Kanske et al., 2015*). Here, we calculated participants' error rates during the ToM questions after the video, collapsed across neutral and negative conditions.

## Statistical analysis

Analysis was performed using SurfStat for Matlab (*Worsley et al., 2009*). We employed linear mixed-effects models, a flexible statistical technique that allows for inclusion of multiple measurements per subjects and irregular measurement intervals (*Pinheiro and Bates, 2000*). In all models, we controlled for age and sex, and random effect of subject. Inference was performed on subject-specific eccentricity/gradient change maps, Δeccentricity/gradient, which were generated by subtracting parcel-wise eccentricity/gradient maps of subsequent time points for a given participant. Main analysis contrasts the TMs in TC1 and TC2 against each other to account for possible training general effects whilst controlling for subject as random effect in the linear model, and age and sex. In follow-up analyses, we compared TMs against retest control cohorts, active control cohort (TC3), and for training cohorts independently.

a) Assessing TM-specific change. To compare two TMs, we compared one TM against another (for example *Affect* versus *Perspective*). To compare a given TM against RCC, we estimated contrasts for TM change relative to RCC (*Presence, Perspective, Affect*).

$$M = 1 + A + S + TM + random(Subject) + I$$

b) Correction for multiple comparisons. Findings were corrected for number of tests within the analysis step using FDR correction (*Benjamini and Hochberg, 1995*).

Theoretically, the cross-over design of the study and the inclusion of number of scans since baseline as covariance controlled for test-retest effects on motion (as participants may become calmer in scanner after repeated sessions). Nevertheless, to control for outliers, we removed all individuals with >0.3 mm/degree movement (*Power et al., 2012*).

## Behavioral prediction

We adopted a supervised framework with cross-validation to predict behavioral change based on change in functional and microstructural organization within five *a-priori* functional networks. We

aimed at predicting attention, compassion, and perspective-taking (*Figure 5*). Before running our model, we regressed out age and sex from the brain markers within the cross-validation loop to avoid leakage. We utilized fivefold cross-validation separating training and test data and repeated this procedure 100 times with different sets of training and test data to avoid bias for separating subjects. Following we performed an elastic net cross-validation procedure with alphas varying from 0.0001 to 1, ratio 1.0, making it a lasso regression. We used sequential feature selection to determine the top 20% of features based on mean absolute error without cross validation. Linear regression for predicting behavioral scores was constructed using the selected features as independent variables within the training data (4/5 segments) and it was applied to the test data (1/5 segment) to predict their behavioral scores. The prediction accuracy was assessed by calculating Pearson's correlation between the actual and predicted behavioral scores as well as their negative mean absolute error, nMAE. To further assess specificity of the behavioral prediction models, we evaluated Pearson's correlation between actual and predicted scores based on randomized scores, as well as using the model to predict out-of-TM data (e.g. for attention in *Presence*; compassion in *Affect* and ToM in *Perspective*; for compassion in *Affect*, attention in *Presence* and ToM in *Perspective*; for ToM in *Perspective*; attention in *Presence* and compassion in *Affect*).

## Data and code availability

In line with EU data regulations (General Data Protection Regulation, GDPR), we regret that data cannot be shared publicly because we did not obtain explicit participant agreement for data-sharing with third parties. Our work is based on personal data (age, sex, and neuroimaging data) that could be matched to individuals. The data is therefore pseudonominized rather than anonymized and falls under the GDPR. Data are available upon request (contact via valk@cbs.mpg.de). Summary data and analysis scripts (Matlab and python) to reproduce primary analyses and figures are publicly available on GitHub (https://github.com/CNG-LAB/social_function_structure_change), and raw data-plots are provided for network-level analyses. To construct gradients, we used the brainspace package, available at brainspace.readthedocs.io. To construct intra-cortical myelin profiles code is available at micapipe.readthedocs.io. Meta-analytical functional MRI maps are downloaded from neurosynth.org and available on GitHub.

## Acknowledgements

Data for the ReSource project were collected between 2013 and 2016 at the Department of Social Neuroscience at the Max Planck Institute for Human Cognitive and Brain Sciences Leipzig. Tania Singer (Principal Investigator) received funding for the ReSource Project from the European Research Council (ERC) under the European Community's Seventh Framework Program (FP7/2007–2013) ERC grant agreement number 205557. Sofie Valk received support from the Max Planck Society (Otto Hahn Award). Boris Bernhardt acknowledges research support from the NSERC (Discovery-1304413), the Canadian Institutes of Health Research (CIHR FDN-154298), SickKids Foundation (NI17-039), Azrieli Center for Autism Research (ACAR-TACC), and the Tier-2 Canada Research Chairs program. Bo-yong Park was funded by the National Research Foundation of Korea (NRF-2021R1F1A1052303; NRF-2022R1A5A7033499), Institute for Information and Communications Technology Planning and Evaluation (IITP) funded by the Korea Government (MSIT) (No. 2022-0-00448, Deep Total Recall: Continual Learning for Human-Like Recall of Artificial Neural Networks; No. RS-2022-00155915, Artificial Intelligence Convergence Innovation Human Resources Development (Inha University); No. 2021-0-02068, Artificial Intelligence Innovation Hub), and Institute for Basic Science (IBS-R015-D1). We thank Lisa Feldman Barrett for fruitful discussions during the conception of this manuscript.

# Additional information

## Funding

| Funder | Grant reference number | Author |
| --- | --- | --- |
| European Research Council | 205557 | Tania Singer |
| Natural Sciences and Engineering Research Council of Canada | Discovery-1304413 | Boris C Bernhardt |
| Canadian Institutes of Health Research | CIHR FDN-154298 | Boris C Bernhardt |
| Sick Kids Foundation | NI17-039 | Boris C Bernhardt |
| Azrieli Foundation | ACAR-TACC | Boris C Bernhardt |
| Canada Research Chairs | Tier 2 | Boris C Bernhardt |
| National Research Foundation of Korea | NRF-2021R1F1A1052303 | Bo-yong Park |
| Institute for Information and Communications Technology Planning and Evaluation (IITP) funded by the Korea Government (MSIT) | No. 2022-0-00448, Deep Total Recall: Continual Learning for Human-Like Recall of Artificial Neural Networks | Bo-yong Park |
| Institute for Basic Science | Institute for Basic Science | Bo-yong Park |
| National Research Foundation of Korea | NRF-2022R1A5A7033499 | Bo-yong Park |
| Institute for Information and Communications Technology Planning and Evaluation (IITP) funded by the Korea Government (MSIT) | No. RS-2022-00155915, Artificial Intelligence Convergence Innovation Human Resources Development (Inha University) | Bo-yong Park |
| Institute for Information and Communications Technology Planning and Evaluation (IITP) funded by the Korea Government (MSIT) | No. 2021-0-02068, Artificial Intelligence Innovation Hub | Bo-yong Park |

The funders had no role in study design, data collection and interpretation, or the decision to submit the work for publication. Open access funding provided by Max Planck Society.

## Author contributions

Sofie Louise Valk, Conceptualization, Data curation, Formal analysis, Investigation, Visualization, Methodology, Writing - original draft; Philipp Kanske, Anne Böckler, Fynn-Mathis Trautwein, Formal analysis, Investigation, Methodology, Writing – review and editing; Bo-yong Park, Seok-Jun Hong, Methodology, Writing – review and editing; Boris C Bernhardt, Conceptualization, Formal analysis, Supervision, Investigation, Methodology, Writing – review and editing; Tania Singer, Conceptualization, Resources, Data curation, Supervision, Funding acquisition, Project administration, Writing – review and editing

## Author ORCIDs

Sofie Louise Valk ⓘ http://orcid.org/0000-0003-2998-6849
Philipp Kanske ⓘ http://orcid.org/0000-0003-2027-8782
Bo-yong Park ⓘ http://orcid.org/0000-0001-7096-337X
Seok-Jun Hong ⓘ http://orcid.org/0000-0002-1847-578X
Fynn-Mathis Trautwein ⓘ http://orcid.org/0000-0001-9928-0193
Boris C Bernhardt ⓘ http://orcid.org/0000-0001-9256-6041

### Ethics

The study was registered at ClinicalTrials.gov under the title.

Human subjects: All participants gave written informed consent and the study was approved by the Research Ethics Committees of the University of Leipzig (#376/12-ff) and Humboldt University in Berlin (#2013-02, 2013-29, 2014-10).

### Decision letter and Author response

Decision letter https://doi.org/10.7554/eLife.85188.sa1
Author response https://doi.org/10.7554/eLife.85188.sa2

## Additional files

### Supplementary files

• Supplementary file 1. Complete summary of supplementary tables and analysis. (a)**.** Descriptive statistics *Presence* (b). Descriptive statistics *Affect* (c). Descriptive statistics *Affect*, excluding active controls (TC3) (d). Descriptive statistics *Perspective* (e). Descriptive statistics Retest controls. (f). Functional eccentricity changes GSR controlled per *a-priori* network. T-values and p-values below $P<0.05$, * indicates FDRp <0.05. (g). Functional eccentricity changes in training cohort 1 per *a-priori* network. T-values and p-values below $P<0.05$, * indicates FDRp <0.05. (h). Functional eccentricity changes in training cohort 2 per *a-priori* network. T-values and p-values below $P<0.05$, * indicates FDRp <0.05. (i). Functional eccentricity changes per *a-priori* network baseline to T1. T-values and p-values below $P<0.05$, * indicates FDRp <0.05. (j). Functional eccentricity changes per *a-priori* network T1 to T3. T-values and p-values below $P<0.05$, * indicates FDRp <0.05. (k). G1-G3 change per *a-priori* network *Presence* vs *Perspective*. T-values and p-values below $P<0.05$, * indicates FDRp <0.05. (l). G1-G3 change per *a-priori* network *Presence* vs *Affect*. T-values and p-values below $P<0.05$, * indicates FDRp <0.05. (m). G1-G3 change per *a-priori* network *Perspective* vs *Affect*. T-values and p-values below $P<0.05$, * indicates FDRp <0.05. (n). Functional eccentricity changes per *a-priori* network controlling for cortical thickness change. T-values and p-values below $P<0.05$, * indicates FDRp <0.05. (o). Functional eccentricity changes per *a-priori* network from baseline to T3. T-values and p-values below $P<0.05$, * indicates FDRp <0.05. (p). Depth-dependent qT1 change per *a-priori* network Training vs Retest Control. T-values and p-values below $P<0.05$. * indicates FDRp <0.05.(q). Descriptives of retest-control change (mean change over T0-T1; T1-T2; T2-T3) as a function of depth-dependent qT1. (r). Descriptive of *Presence* change (mean change over T0-T1) as a function of depth-dependent qT1 (s). Descriptives of *Affect* change (mean change over T0-T1, T1-T2 and T2-T3) as a function of depth-dependent qT1. (t). Descriptives of *Affect* change (mean change over T1-T2 and T2-T3) as a function of depth-dependent qT1. (u). Descriptives of *Perspective* change (mean change over T1-T2 and T2-T3) as a function of depth-dependent qT1. (v). Depth-dependent qT1 change per *a-priori* network *Presence* vs *Perspective*. T-values and p-values below $P<0.05$, and Cohen's D effect size, * indicates FDRp <0.05. (w). Depth-dependent qT1 change per *a-priori* network *Presence* vs *Affect*. T-values and p-values below $P<0.05$, and Cohen's D effect size, * indicates FDRp <0.05. (x). Depth-dependent qT1 change per *a-priori* network *Perspective* vs *Affect*. T-values and p-values below $P<0.05$, and Cohen's D effect size, * indicates FDRp <0.05. (y). Depth-dependent qT1 change per *a-priori* network *Presence* vs *Perspective* in TC1. T-values and p-values below $P<0.05$. * indicates FDRp <0.05. (z). Depth-dependent qT1 change per *a-priori* network *Presence* vs *Affect* in TC1. T-values and p-values below $P<0.05$. * indicates FDRp <0.05. (za). Depth-dependent qT1 change per *a-priori* network *Perspective* vs *Affect* in TC1. T-values and p-values below $P<0.05$. * indicates FDRp <0.05. (zb). Depth-dependent qT1 change per *a-priori* network *Presence* vs *Perspective* in TC2. T-values and p-values below $P<0.05$. * indicates FDRp <0.05. (zc). Depth-dependent qT1 change per *a-priori* network *Presence* vs *Affect* in TC2. T-values and p-values below $P<0.05$. * indicates FDRp <0.05. (zd). Depth-dependent qT1 change per *a-priori* network *Perspective* vs *Affect* in TC2. T-values and p-values below $P<0.05$. * indicates FDRp <0.05. (ze). Depth-dependent qT1 change per *a-priori* network baseline – T1: TC1 (*Presence*) versus Retest Control. T-values and p-values below $P<0.05$. * indicates FDRp <0.05. (zf). Depth-dependent qT1 change per *a-priori* network baseline – T1: TC2 (*Presence*) versus Retest Control. T-values and p-values below $P<0.05$. * indicates FDRp <0.05. (zg). Depth-dependent qT1 change per *a-priori* network baseline – T1: *Affect* TC3 vs Retest Control. T-values and p-values below $P<0.05$. * indicates FDRp <0.05. (zh). Depth-dependent qT1 change per *a-priori* network baseline – T1: *Presence* TC1 vs *Affect* TC3. T-values and p-values below $P<0.05$. * indicates FDRp <0.05. (zi). Depth-dependent qT1 change per *a-priori* network baseline – T1: *Presence* TC2 vs *Affect* TC3.

T-values and p-values below $P<0.05$. * indicates FDRp $<0.05$. (zj). Depth-dependent qT1 change per *a-priori* network T1 – T3: *Perspective* versus *Affect* (TC1 +TC2). T-values and p-values below $P<0.05$. * indicates FDRp $<0.05$. (zk). Depth-dependent qT1 change per *a-priori* network T1 – T3: *Affect* vs Retest Control. * T-values and p-values below $P<0.05$. * indicates FDRp $<0.05$. (zl). Depth-dependent qT1 change per *a-priori* network T1 – T3: *Perspective* vs Retest Control. T-values and p-values below $P<0.05$. * indicates FDRp $<0.05$. (zm). Depth-dependent qT1 change per *a-priori* network controlling for CTX: *Presence* versus *Perspective*. T-values and p-values below $P<0.05$. * indicates FDRp $<0.05$.(zn). Depth-dependent qT1 change per *a-priori* network controlling for CTX: *Presence* versus *Affect*. T-values and p-values below $P<0.05$. * indicates FDRp $<0.05$. (zo). Depth-dependent qT1 change per *a-priori* network controlling for CTX: *Perspective* versus *Affect*. T-values and p-values below $P<0.05$. * indicates FDRp $<0.05$.

- MDAR checklist

### Data availability

In line with EU data regulations (General Data Protection Regulation, GDPR), we regret that data cannot be shared publicly because we did not obtain explicit participant agreement for data-sharing with third parties. Our work is based on personal data (age, sex, and neuroimaging data) that could be matched to individuals. The data is therefore pseudonominized rather than anonymized and falls under the GDPR. Data are available upon request (contact via valk@cbs.mpg.devalk@cbs.mpg.de). Summary data and analysis scripts (Matlab and python) to reproduce primary analyses and figures are publicly available on GitHub (https://github.com/CNG-LAB/social_function_structure_change copy archived at *cng-lab, 2023*), and raw data-plots are provided for network-level analyses. To construct gradients, we used the brainspace package, available at https://brainspace.readthedocs.io/en/latest/. To construct intra-cortical myelin profiles code is available at https://micapipe.readthedocs.io/en/latest/. Meta-analytical functional MRI maps are downloaded from https://neurosynth.org/ and available on GitHub.

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
