## [Editor Report]

This important work extensively quantifies changes in cortical hierarchical organization induced by different types of social cognitive training. The evidence supporting this is compelling: the authors employ rigorous and complementary multi-modal neuroimaging assessments in a very large sample, measuring longitudinal changes in functional and structural metrics of cortical hierarchical organization. This work has broad applicability to basic neuroscience, illuminating the link between anatomical and functional hierarchies in the brain and social skills, and is also of interest to clinical psychology audiences due to its relevance to interventions such as mindfulness-based therapies.

---

## [Decision Letter]

**Decision letter after peer review:**

[Editors’ note: the authors submitted for reconsideration following the decision after peer review. What follows is the decision letter after the first round of review.]

Thank you for submitting the paper "Changing the social brain: plasticity along macro-scale axes of functional connectivity following social mental training" for consideration by *eLife*. Your article has been reviewed by 3 expert peer reviewers, and the evaluation has been overseen by Drs. Shackman (Reviewing Editor) and Makin (Senior Editor).

I am sorry to say that---after consultation with the reviewers---we have collectively decided that this work will not be considered further for publication by *eLife*. In brief, while there was a lot to like about your study (as we highlight in the public summary below), upon consultation the reviewers agreed that the conceptual and practical significance of the work is incremental.

*Reviewer #1:*

Valk et al. report a thorough, causal assessment of the brain systems involved in human social behavior. Behavioral and neuroscience research has shown that social cognition is supported by multiple distinct socio-affective, socio-cognitive, and attentional components. Prior human neuroimaging studies have demonstrated that these components are associated with activity in distributed cortical systems, including portions of the default mode, ventral attention, and multiple demand networks. The present work uses behavioral training over 9 months to causally test the relationship between component processes and large-scale cortical systems involved in social cognition. The authors find that socio-cognitive training increased functional integration of default and multiple demand regions whereas attention-mindfulness resulted in functional segregation. They further adopt a machine learning framework to show that changes in functional organization modestly predict changes in behavioral performance during training. These data provide one of the most thorough evaluations of brain systems involved in social cognition to date.

Although the main conclusions of the paper are generally supported by the data, aspects of the analysis and reporting weaken the manuscript:

– A related paper from this group (Valk, S. L., et al. 2017. Science Advances) reported changes in structural plasticity (cortical thickness) following behavioral training in the same cohort. It appears that several areas in the temporal lobe show both structural changes and functional network reorganization following training. It is not clear how the functional changes identified in the present report relate to the previously identified structural alterations.

– Many of the results appear to focus on the statistical significance of simple effects for the individual training modules (e.g., reporting t-statistics and associated p-values for changes in the eccentricity of brain region/networks following a single module). If the reported results do reflect contrasts between training modules (as much is stated in the methods), it is not clear to what extent the modules (and the active control) differ from one another because descriptive statistics and effect sizes are not reported. This does not appear to be a critical issue for some of the main results (where the same networks/regions have opposite effects), but it makes it difficult to evaluate the strength of the findings as a whole.

– Although potentially quite interesting, it is not clear that the connectivity-based prediction of behavioral changes is very robust. Effect sizes are small to medium, the methods used for these analyses are prone to data leakage (and steps to protect against these problems are not described), effect size estimates are based on cross-validation alone (as opposed to out-of-sample tests), and there are many experimenter degrees of freedom.

Additional specific recommendations:

– In addition to reporting t-statistics and p values, report descriptive statistics (mean, standard deviation, and confidence intervals) for key comparisons.

– Assess (or control for) the effect of cortical thickness on functional network reorganization. Based on the authors' past work, it appears that they expect these measures to be related, but they are not evaluated or even discussed in the present paper.

– Improve the description of the machine learning approach. Was dimension reduction performed? Based on the main text it appears PCA may have been performed, but this is not discussed in the methods. If so, how was the number of components selected? Was this done on each cross-validation fold? In the absence of these details, the reader is left to assume that the reported effects could be the result of overfitting.

– Because that cross-validation can lead to overly optimistic performance estimates, it would be helpful to perform permutation-based inference against "chance" levels of performance. Multi-level block permutation is recommended given the hierarchical nature of this dataset (see Winkler, A. M., Webster, M. A., Vidaurre, D., Nichols, T. E., & Smith, S. M. (2015). Multi-level block permutation. Neuroimage, 123, 253-268.)

– The word 'module' is used to describe the different behavioral training regimens that participants completed. As the authors are likely aware, this word is also commonly used to describe network structure. This makes phrases such as "module-specific behavioral change" and "behavioral changes across modules" difficult to parse. The authors may want to consider revising their use of the term module.

*Reviewer #2:*

The study has several strengths:

– Capitalizes on an extensive training program (across 9 months) in a sample of participants that has good size.

– Leverages several advanced analysis methods to characterize how large-scale organization of functional connectivity is altered by training.

– Results show that changes are observed along major axes of functional connectivity organization. Notably, these changes were correlated with behavioral changes, although the associations were rather modest in size.

The study also has several weaknesses that undermine significance:

– The paper has a large number of analyses and results. However, it not entirely clear how the study advances knowledge except in a general fashion that "functional connectivity" changes.

– The central goal of the study is unclear. If the objective was to, as stated in some places, determine changes in integration/segregation of networks, the approach seems too indirect. A direct approach would evaluate these properties with graph theory. The authors do provide some results in that direction, but it is not clear why the results are secondary and mainly used to lend support to their gradients approach. Observing a correlation of r>0.5 is provided as supporting evidence but only in a very general analysis, not specific instances.

– The paper has a large number of analyses and several processing choices were made. While many appear reasonable, several choices are potentially problematic. For example, participants with gradients that correlated less than 0.5 with the average were discarded. The corrections for multiple comparisons were not clearly justified. In one case, clusters were accepted if p less than 0.005 and in another 0.01. More generally, if the correction was applied, why not adopt 0.05?

– Several analyses were performed but almost completely relegated to supplementary material. It is unclear how the Neurosynth and genetic analyses contribute to the study. Their inclusion contributes to the impression that the authors decided to try several analyses and see what relationships would be observed.

Additional specific recommendations:

– The way the paper is presented, it appears that the authors wanted to analyze the data in terms of the "gradients" approach and use that to investigate questions that would be better addressed with other techniques. If this impression is wrong, the authors would be encouraged to try to motivate the study more clearly.

– The paper is extremely dense and difficult to follow. I would recommend an extensive rewrite.

*Reviewer #3:*

This is a comprehensive and well powered study demonstrating that Presence-training results in increased eccentricity of bilateral temporal and right superior parietal areas, Perspective-training resulted in decreased eccentricity of right temporal regions and insular cortex, and there was no effect of Affect-training. These findings were significant following on family-wise error correction. Whether or not GSR was run, did not significantly affect the results. The sample size is excellent with two training groups (N=80 and N=81), a matched test-retest control cohort (N=90), and a separate single training set (N=81). Subjects were imaged at baseline and across 3 sessions x 3 monthly intervals (4 sessions total).

Specific recommendations

– Taking the eigenvectors of connectivity gradients (G1, G2, G3) the authors calculated the distance from the center of this coordinate system calling this the eccentricity. This measure is presumed to capture the vertex-wise intrinsic functional integration (low eccentricity) and segregation (high eccentricity).

– The section on task-based networks is not all that clearly written (page 8, line 7 onwards). Task-based networks were defined based on task-based fMRI and then these networks were used as input to the spatial gradient calculations. There should be some transition language after describing how each of the task-based networks was calculated, before immediately discussing how the dimensionality of the resting state connectomes was reduced. I had to go back and forth between methods, this section and the supplementary section to figure out if the connectivity matrix was based on the 400 node parcellation (used later for graph theory) or the networks defined by the tasks. It's not explicitly stated.

– While care is taken in the prediction section to randomize and then in other sections to look at the effect of GSR, there is no consideration given to the sensitivity of these results to various thresholding steps. In particular, the task defined networks are arbitrarily thresholded – and it is unclear how sensitive the results are to this threshold.

– Sensitivity of the gradient calculations to how the task-based thresholds were applied to define the networks should be provided. If the task-based maps are simply changing with practice effects then wouldn't this change the gradient calculations because of different inputs or defined networks. I initially assumed task-based networks were defined at each time point – but now I realize perhaps they are only defined once. Were the paradigms for functional localization only run once or were they run each time (before and after training)? We know that task-based modules change with practice effects (and presumably with task-relevant training). So how would changes at the level of these functional subunits influence these results? Is it appropriate to measure these only once and assume they are appropriate for each time point? If the nodes change with training across time would that account for the differences in eccentricity observed? Some supplementary data showing how the task activation maps change with training intervention might be helpful – if the tasks were run more than once.

– Does overall eccentricity predict behavior? It's unclear when it is stated that G1-G3 combined could predict compassion and ToM, if this is the same as using eccentricity to predict these behaviors. Is using the first three components the same as using the single eccentricity measure? If so then why not call it that since the previous section is all focused on eccentricity.

– In the prediction work it's not clear what exact behavioral measures were tested. There is a description on page 19 under behavioral markers starting on line 33, but it would be helpful if this was described earlier (maybe this is a journal formatting problem). However, in this section there is a description of how the measure for compassion and ToM were obtained but no description of the attention score.

– It wasn't clear to me if the specific networks tested did better at predicting the related tasks or if cross-network-task predictions were investigated. For example, you measured the extent to whether G1-G3 in the attention network predicted attention scores. Did these same G1-G3 values predict socio-affective or ToM scores? Did the socio-affective network predict socio-cognitive skills better than attention and ToM? In other words what happened if you mixed and matched these across networks – how critical are the specific network definitions to the prediction?

– Was any consideration given to the robustness of results to the parameter α which controls the influence of the density of sampling points on the manifold and also later the 10% sparsity threshold? How sensitive are these results to those decisions?

[Editors’ note: further revisions were suggested prior to acceptance, as described below.]

Thank you for resubmitting your work entitled "Functional and microstructural plasticity following social and interoceptive mental training." for further consideration by *eLife*. Your revised article has been evaluated by Tamar Makin (Senior Editor) and a Reviewing Editor (Camilla Nord).

The manuscript has been improved but there are some remaining issues that need to be addressed, as outlined below:

The reviewers generally feel your manuscript is significant to the wider field, and with clarification and editing, would be of interest to a broad audience. They highlight several strengths, including the longitudinal approach and sample size. They suggest several key changes that could be made, in particular, two reviewers suggest strengthening the clarity of your central argument (i.e. what was the main goal of the study?), two reviewers suggest tests of the robustness/specificity of your predictive approach (e.g. "does G1-G3 in the attention network predict socio-affective or ToM scores?"), and there are a number of helpful minor clarifications suggested, including adding some of the analytical detail currently included only in the previous/referenced publications to the manuscript itself. We agree these changes would be useful for the clarity and completeness of the manuscript.

*Reviewer #1 (Recommendations for the authors):*

This study has a number of strengths, including sample size, longitudinal experimental design, and MRI methods. The connections between depth-specific qT1 and resting state network features, as well as their combined ability to predict performance after cognitive training are particularly interesting. Overall, the study provides a thorough examination of training-related functional and structural changes in social-cognitive related brain networks. While the authors' individual claims are supported by their data, the main conclusion of the manuscript seems diluted as the many findings and interpretations presented are not tightly connected.

There are a few areas in which the methodology/rationale is unclear as described and could benefit from further refinement. These are detailed below.

1. The manuscript presents a set of interesting observations but seems to be missing a main argument. There can be several potential candidates for it, including (1) social cognitive training leading to changes in resting-state connectivity and depth-dependent qT1, indicating functional and microstructural plasticity, or (2) affect, presence, and perspective training leading to differential changes in network integration/segregation, supporting the global workspace theory of cognition, or (3) training-induced functional and intracortical microstructural changes are related depending on the cortical depth and specific training task, and so on. This is a very information-heavy study, so I would recommend distilling a more specific main idea and tailoring your results/discussion to it.

2. The data are analyzed and presented on multiple levels of granularity, which is very impressive but in the meantime makes the Results section hard to follow. For example, the findings for longitudinal connectivity changes are discussed in terms of (1) eccentricity of the whole brain, (2) each of the three gradients, (3) a priori functional networks, and combinations of the above (e.g., changes along gradient 3 of the a priori attention network). I would recommend the authors trim any redundant analyses or clarify the distinction between these analytical levels and the significance of including all.

3. Following the previous comment, it would be helpful to briefly introduce the importance of extracting and examining three gradients, especially G3, which in my experience is less seen in the literature.

4. Please provide more details for statistical analysis in the methods section. For example, in the comparison of presence vs. perspective, is it overall changes after presence training vs. after perspective training, or changes in the selective group that had perspective training after presence?

5. 12 cortical layers were sampled from the qT1 map acquired at 1mm isotropic resolution. Given that many changes were observed near the pial surface, such as Figure 3A-ii. Please discuss if and how the effects of partial volume averaging were accounted for. Were all qT1-related analyses controlled for cortical thickness?

6. I must admit that I'm a bit confused about the qT1 layer results, especially those in Figure 3B. The colored lines seem to indicate across-depth qT1 changes in each functional region. Then why would there be four lines for all networks? For example, why is there a green line, indicating perspective network, in the plot for attention network results? More details in the figure caption would be helpful.

*Reviewer #2 (Recommendations for the authors):*

In the manuscript entitled "Functional and microstructural plasticity following social and interoceptive mental training" by Valk et al., the authors present an analysis of the ReSource study data wherein they examine how gradients of functional connectivity and microstructure change following mindfulness and cognitive training. Their findings indicate that individuals' functional connectivity and cortical microstructure change longitudinally in response to these interventions, providing compelling evidence that interventions such as mindfulness change the brain's structure. Additionally, the authors used these changes to predict measures of attention, compassion, and perspective taking.

I was invited to review this manuscript after an initial round of revisions. However, I was not provided a manuscript with changes tracked, which meant I could not review the paper before and after the authors' changes. Additionally, the authors provided only a summary of their changes to the original reviewers, rather than a point-by-point response. This made it unclear which of the original reviewers' comments were addressed and which were not. As such, instead of offering new comments, I have chosen to go through the original reviewers' comments and pick out ones that I (i) think are important and (ii) do not believe have been addressed by the authors.

"The word 'module' is used to describe the different behavioral training regimens that participants completed. As the authors are likely aware, this word is also commonly used to describe network structure. This makes phrases such as "module-specific behavioral change" and "behavioral changes across modules" difficult to parse. The authors may want to consider revising their use of the term module."

I agree with R1 here. The use of the term 'module' conflicts with the broader fields typical use of the term (referring to modules within graph topology). Could the authors consider rephrasing to 'training module' or TM?

"The central goal of the study is unclear. If the objective was to, as stated in some places, determine changes in integration/segregation of networks, the approach seems too indirect. A direct approach would evaluate these properties with graph theory. The authors do provide some results in that direction, but it is not clear why the results are secondary and mainly used to lend support to their gradients approach."

I agree with R2 here. In general, I found the link between the authors measure of eccentricity and the twin pillars of functional integration and segregation to be unclear. The authors state:

"we calculated region-wise distances to the center of a coordinate system formed by the first three gradients G1, G2, and G3 for each individual [based on the Schaefer 400 parcellation (67)]. Such a gradient eccentricity measures captures intrinsic functional integration (low eccentricity) vs segregation (high eccentricity) in a single scalar value (68)."

I understand that this statement includes a relevant citation, but I think there is room for more intuition building here. What does high eccentricity correspond to in this distance calculation? Is it high distance from the center of a coordinate system? If so, why does high distance from center correspond to a segregated brain? Why does low distance from center correspond to an integrated brain? Like R2, I found it difficult to reconcile this gradient-distance based metric with my graph topology understanding of segregation and integration.

Prediction analyses:

R1 stated:

"Although potentially quite interesting, it is not clear that the connectivity-based prediction of behavioral changes is very robust. Effect sizes are small to medium, the methods used for these analyses are prone to data leakage (and steps to protect against these problems are not described), effect size estimates are based on cross-validation alone (as opposed to out-of-sample tests), and there are many experimenter degrees of freedom."

R3 stated:

"In the prediction work it's not clear what exact behavioral measures were tested. There is a description on page 19 under behavioral markers starting on line 33, but it would be helpful if this was described earlier (maybe this is a journal formatting problem). However, in this section there is a description of how the measure for compassion and ToM were obtained but no description of the attention score."

And

"It wasn't clear to me if the specific networks tested did better at predicting the related tasks or if cross-network-task predictions were investigated. For example, you measured the extent to whether G1-G3 in the attention network predicted attention scores. Did these same G1-G3 values predict socio-affective or ToM scores? Did the socio-affective network predict socio- cognitive skills better than attention and ToM? In other words what happened if you mixed and matched these across networks – how critical are the specific network definitions to the prediction?"

Presently, I do not think these comments about the authors' prediction analyses have been addressed. Moreover, I too have concerns about the authors' approach. For example, the authors state "Before running our model, we regressed out age and sex from the brain markers." Here, it's unclear whether this nuisance regression was done in a leakage resistant way or not. This sentence implies that age and sex were regressed out of all the data in a single step prior to running the prediction model. If this is the case, this approach will cause leakage and spuriously boost prediction performance. To avoid this, the authors should consider incorporating nuisance regression into their cross-validation model, wherein nuisance models are fit to the training data and applied to the test data.

*Reviewer #3 (Recommendations for the authors):*

The study by Valk and colleagues investigated the impact of various forms of social mental training on resting-state functional connectivity and myeloarchitecture using a large-scale longitudinal multimodal dataset. The examination of these changes in combination is particularly noteworthy.

However, a few key points should be addressed to further improve the study.

[Line 140] The use of task-based networks to summarize changes across the cortex may be problematic, as the metrics averaged within each network may be influenced by small clusters, rather than reflecting the entire network. The FDR-survived changes in Attention and Interoception networks may be due to overlap in vertices, rather than network properties. To avoid this, it may be beneficial to consider alternative methods for summarizing whole-cortex changes.

[Line 217] The motivation for the selective analysis is clear, but the overall effects of the training were unclear until later in the manuscript (Line 234 and Supplementary Figure 6). It may be beneficial to describe the overall effects of training upfront.

[Line 268] The predictive performance of the models was not explicitly tested to determine if it was above the chance level. For a nonparametric test, it may be useful to calculate the null distribution of chance level performance through the use of permuted pairs of predictors and targets.

---

## [Author Response]

[Editors’ note: the authors resubmitted a revised version of the paper for consideration. What follows is the authors’ response to the first round of review.]

Reviewer #1:Valk et al. report a thorough, causal assessment of the brain systems involved in human social behavior. Behavioral and neuroscience research has shown that social cognition is supported by multiple distinct socio-affective, socio-cognitive, and attentional components. Prior human neuroimaging studies have demonstrated that these components are associated with activity in distributed cortical systems, including portions of the default mode, ventral attention, and multiple demand networks. The present work uses behavioral training over 9 months to causally test the relationship between component processes and large-scale cortical systems involved in social cognition. The authors find that socio-cognitive training increased functional integration of default and multiple demand regions whereas attention-mindfulness resulted in functional segregation. They further adopt a machine learning framework to show that changes in functional organization modestly predict changes in behavioral performance during training. These data provide one of the most thorough evaluations of brain systems involved in social cognition to date.Although the main conclusions of the paper are generally supported by the data, aspects of the analysis and reporting weaken the manuscript:– A related paper from this group (Valk, S. L., et al. 2017. Science Advances) reported changes in structural plasticity (cortical thickness) following behavioral training in the same cohort. It appears that several areas in the temporal lobe show both structural changes and functional network reorganization following training. It is not clear how the functional changes identified in the present report relate to the previously identified structural alterations.– Many of the results appear to focus on the statistical significance of simple effects for the individual training modules (e.g., reporting t-statistics and associated p-values for changes in the eccentricity of brain region/networks following a single module). If the reported results do reflect contrasts between training modules (as much is stated in the methods), it is not clear to what extent the modules (and the active control) differ from one another because descriptive statistics and effect sizes are not reported. This does not appear to be a critical issue for some of the main results (where the same networks/regions have opposite effects), but it makes it difficult to evaluate the strength of the findings as a whole.– Although potentially quite interesting, it is not clear that the connectivity-based prediction of behavioral changes is very robust. Effect sizes are small to medium, the methods used for these analyses are prone to data leakage (and steps to protect against these problems are not described), effect size estimates are based on cross-validation alone (as opposed to out-of-sample tests), and there are many experimenter degrees of freedom.Additional specific recommendations:– In addition to reporting t-statistics and p values, report descriptive statistics (mean, standard deviation, and confidence intervals) for key comparisons.– Assess (or control for) the effect of cortical thickness on functional network reorganization. Based on the authors' past work, it appears that they expect these measures to be related, but they are not evaluated or even discussed in the present paper.– Improve the description of the machine learning approach. Was dimension reduction performed? Based on the main text it appears PCA may have been performed, but this is not discussed in the methods. If so, how was the number of components selected? Was this done on each cross-validation fold? In the absence of these details, the reader is left to assume that the reported effects could be the result of overfitting.– Because that cross-validation can lead to overly optimistic performance estimates, it would be helpful to perform permutation-based inference against "chance" levels of performance. Multi-level block permutation is recommended given the hierarchical nature of this dataset (see Winkler, A. M., Webster, M. A., Vidaurre, D., Nichols, T. E., & Smith, S. M. (2015). Multi-level block permutation. Neuroimage, 123, 253-268.)– The word 'module' is used to describe the different behavioral training regimens that participants completed. As the authors are likely aware, this word is also commonly used to describe network structure. This makes phrases such as "module-specific behavioral change" and "behavioral changes across modules" difficult to parse. The authors may want to consider revising their use of the term module.Reviewer #2:The study has several strengths:– Capitalizes on an extensive training program (across 9 months) in a sample of participants that has good size.– Leverages several advanced analysis methods to characterize how large-scale organization of functional connectivity is altered by training.– Results show that changes are observed along major axes of functional connectivity organization. Notably, these changes were correlated with behavioral changes, although the associations were rather modest in size.The study also has several weaknesses that undermine significance:– The paper has a large number of analyses and results. However, it not entirely clear how the study advances knowledge except in a general fashion that "functional connectivity" changes.– The central goal of the study is unclear. If the objective was to, as stated in some places, determine changes in integration/segregation of networks, the approach seems too indirect. A direct approach would evaluate these properties with graph theory. The authors do provide some results in that direction, but it is not clear why the results are secondary and mainly used to lend support to their gradients approach. Observing a correlation of r>0.5 is provided as supporting evidence but only in a very general analysis, not specific instances.– The paper has a large number of analyses and several processing choices were made. While many appear reasonable, several choices are potentially problematic. For example, participants with gradients that correlated less than 0.5 with the average were discarded. The corrections for multiple comparisons were not clearly justified. In one case, clusters were accepted if p less than 0.005 and in another 0.01. More generally, if the correction was applied, why not adopt 0.05?– Several analyses were performed but almost completely relegated to supplementary material. It is unclear how the Neurosynth and genetic analyses contribute to the study. Their inclusion contributes to the impression that the authors decided to try several analyses and see what relationships would be observed.Additional specific recommendations:– The way the paper is presented, it appears that the authors wanted to analyze the data in terms of the "gradients" approach and use that to investigate questions that would be better addressed with other techniques. If this impression is wrong, the authors would be encouraged to try to motivate the study more clearly.– The paper is extremely dense and difficult to follow. I would recommend an extensive rewrite.Reviewer #3:This is a comprehensive and well powered study demonstrating that Presence-training results in increased eccentricity of bilateral temporal and right superior parietal areas, Perspective-training resulted in decreased eccentricity of right temporal regions and insular cortex, and there was no effect of Affect-training. These findings were significant following on family-wise error correction. Whether or not GSR was run, did not significantly affect the results. The sample size is excellent with two training groups (N=80 and N=81), a matched test-retest control cohort (N=90), and a separate single training set (N=81). Subjects were imaged at baseline and across 3 sessions x 3 monthly intervals (4 sessions total).Specific recommendations– Taking the eigenvectors of connectivity gradients (G1, G2, G3) the authors calculated the distance from the center of this coordinate system calling this the eccentricity. This measure is presumed to capture the vertex-wise intrinsic functional integration (low eccentricity) and segregation (high eccentricity).– The section on task-based networks is not all that clearly written (page 8, line 7 onwards). Task-based networks were defined based on task-based fMRI and then these networks were used as input to the spatial gradient calculations. There should be some transition language after describing how each of the task-based networks was calculated, before immediately discussing how the dimensionality of the resting state connectomes was reduced. I had to go back and forth between methods, this section and the supplementary section to figure out if the connectivity matrix was based on the 400 node parcellation (used later for graph theory) or the networks defined by the tasks. It's not explicitly stated.– While care is taken in the prediction section to randomize and then in other sections to look at the effect of GSR, there is no consideration given to the sensitivity of these results to various thresholding steps. In particular, the task defined networks are arbitrarily thresholded – and it is unclear how sensitive the results are to this threshold.– Sensitivity of the gradient calculations to how the task-based thresholds were applied to define the networks should be provided. If the task-based maps are simply changing with practice effects then wouldn't this change the gradient calculations because of different inputs or defined networks. I initially assumed task-based networks were defined at each time point – but now I realize perhaps they are only defined once. Were the paradigms for functional localization only run once or were they run each time (before and after training)? We know that task-based modules change with practice effects (and presumably with task-relevant training). So how would changes at the level of these functional subunits influence these results? Is it appropriate to measure these only once and assume they are appropriate for each time point? If the nodes change with training across time would that account for the differences in eccentricity observed? Some supplementary data showing how the task activation maps change with training intervention might be helpful – if the tasks were run more than once.– Does overall eccentricity predict behavior? It's unclear when it is stated that G1-G3 combined could predict compassion and ToM, if this is the same as using eccentricity to predict these behaviors. Is using the first three components the same as using the single eccentricity measure? If so then why not call it that since the previous section is all focused on eccentricity.– In the prediction work it's not clear what exact behavioral measures were tested. There is a description on page 19 under behavioral markers starting on line 33, but it would be helpful if this was described earlier (maybe this is a journal formatting problem). However, in this section there is a description of how the measure for compassion and ToM were obtained but no description of the attention score.– It wasn't clear to me if the specific networks tested did better at predicting the related tasks or if cross-network-task predictions were investigated. For example, you measured the extent to whether G1-G3 in the attention network predicted attention scores. Did these same G1-G3 values predict socio-affective or ToM scores? Did the socio-affective network predict socio-cognitive skills better than attention and ToM? In other words what happened if you mixed and matched these across networks – how critical are the specific network definitions to the prediction?– Was any consideration given to the robustness of results to the parameter α which controls the influence of the density of sampling points on the manifold and also later the 10% sparsity threshold? How sensitive are these results to those decisions?

We thank the editor and Reviewers for considering out work. Following this feedback, we have taken a step back on the project and reformulated our research question and goals, focusing in biological relevance. Therefore, in a new paper, we have now integrated quantitative T1 relaxometry data based on notions of overlap between microstructural profiles of regions and its function (1-5). Moreover, to increase clarity, we have now put main focus on meta-analytical functional networks matching the functional processes hypothesized to be targeted by each of the studies’ training modules. Second, based on specific recommendations of the Reviewers, we have updated our machine learning procedure, provided details on un-contrasted training-specific changes from main contrast, performed a cross-modality evaluation to probe the biological relevance of functional eccentricity changes and provided additional clarification for the rationale and methological steps taken in the work. Thanks to the helpful input, these steps resulted in a novel paper integrating the main insights from the previous work.

[Editors’ note: what follows is the authors’ response to the second round of review.]

The manuscript has been improved but there are some remaining issues that need to be addressed, as outlined below:The reviewers generally feel your manuscript is significant to the wider field, and with clarification and editing, would be of interest to a broad audience. They highlight several strengths, including the longitudinal approach and sample size. They suggest several key changes that could be made, in particular, two reviewers suggest strengthening the clarity of your central argument (i.e. what was the main goal of the study?), two reviewers suggest tests of the robustness/specificity of your predictive approach (e.g. "does G1-G3 in the attention network predict socio-affective or ToM scores?"), and there are a number of helpful minor clarifications suggested, including adding some of the analytical detail currently included only in the previous/referenced publications to the manuscript itself. We agree these changes would be useful for the clarity and completeness of the manuscript.Reviewer #1 (Recommendations for the authors):This study has a number of strengths, including sample size, longitudinal experimental design, and MRI methods. The connections between depth-specific qT1 and resting state network features, as well as their combined ability to predict performance after cognitive training are particularly interesting. Overall, the study provides a thorough examination of training-related functional and structural changes in social-cognitive related brain networks. While the authors' individual claims are supported by their data, the main conclusion of the manuscript seems diluted as the many findings and interpretations presented are not tightly connected.There are a few areas in which the methodology/rationale is unclear as described and could benefit from further refinement. These are detailed below.1. The manuscript presents a set of interesting observations but seems to be missing a main argument. There can be several potential candidates for it, including (1) social cognitive training leading to changes in resting-state connectivity and depth-dependent qT1, indicating functional and microstructural plasticity, or (2) affect, presence, and perspective training leading to differential changes in network integration/segregation, supporting the global workspace theory of cognition, or (3) training-induced functional and intracortical microstructural changes are related depending on the cortical depth and specific training task, and so on. This is a very information-heavy study, so I would recommend distilling a more specific main idea and tailoring your results/discussion to it.

We thank the Reviewer for this comment. While we agree the study is information heavy, our main argument can be summarized and interpreted as follows:

Training of attention-mindfulness together with socio-emotional and socio-cognitive skills leads to differential longitudinal changes in resting-state connectivity. These changes were paralleled by depth-varying microstructural cortical reorganization. As such, our findings are compatible with previous work suggesting a link between anatomical and functional hierarchies and global workspace theories of cognition (1-3).

We now incorporated these arguments in the Abstract and main text.

Abstract:

“The human brain supports social cognitive functions, including Theory of Mind, empathy, and compassion, through its intrinsic hierarchical organization. However, it remains unclear how the learning and refinement of social skills shapes brain function and structure. We studied if different types of social mental training induce changes in cortical function and microstructure, investigating 332 healthy adults (197 women, 20-55 years) with repeated multimodal neuroimaging and behavioral testing. Our neuroimaging approach examined longitudinal changes in cortical functional gradients and myelin-sensitive T1 relaxometry, two complementary measures of cortical hierarchical organization. We observed marked changes in intrinsic cortical function and microstructure, which varied as a function of social training content. In particular, cortical function and microstructure changed as a result of attention-mindfulness and socio-cognitive training in regions functionally associated with attention and interoception, including insular and parietal cortices. Conversely, socio-affective and socio-cognitive training resulted in differential microstructural changes in regions classically implicated in interoceptive and emotional processing, including insular and orbitofrontal areas, but did not result in functional reorganization. Notably, longitudinal changes in cortical function and microstructure predicted behavioral change in attention, compassion and perspective-taking. Our work demonstrates functional and microstructural plasticity after the training of social and interoceptive functions, and illustrates the bidirectional relationship between brain organization and human social skills.”

Introduction:

“Despite the progress in the mapping of the functional topography of networks mediating social and interoceptive abilities, the interplay between social behavior and brain organization is less well understood (4). Prior research has shown that cortical function and microstructure follow parallel spatial patterns, notably a sensory-transmodal axis that may support the differentiation of sensory and motor function from higher order cognitive processes, such as social cognition (5-9). Put differently, a sensory-transmodal framework situates abstract social and interoceptive functions in transmodal anchors, encompassing both heteromodal regions (such as the prefrontal cortex, posterior parietal cortex, lateral temporal cortex, and posterior parahippocampal regions) as well as paralimbic cortices (including orbitofrontal, insular, temporo-polar, cingulate, and parahippocampal regions) (10). Distant from sensory systems, transmodal cortices take on functions that are only loosely constrained by the immediate environment (11), allowing internal representations to contribute to more abstract, and social cognition and emotion (7, 8, 11-18), thereby enhancing behavioral flexibility (13, 19). However, despite the presumed link between cortical microstructure and functional processes it may support, whether and how changes in social behavior impact intrinsic function and microstructure it is not known to date.

Here, we leverage the ReSource study dataset to assess whether the targeted training of attention-interoception, socio-affective, and socio-cognitive skills can lead to domain-specific reorganization of (i) intrinsic function (as indexed by resting-state fMRI connectivity gradient analysis), and (ii) cortical microstructure (as indexed by quantitative T1 relaxometry, probed along the direction of cortical columns (20-22)). Such results would be in line with prior observations suggesting coupled change in brain structure and function (23, 24), and would help to gain insights into the association between social skills and models of brain organization. Longitudinal analyses of subjects-specific measures of functional integration and segregation evaluated whether these changes corresponded to corresponding change in intracortical microstructure. We also tested for associations to behavioral change in attention, compassion, and ToM markers using machine learning with cross-validation, to evaluate behavioral relevance at the individual level.”

Discussion:

“We studied whether targeted training of human (social)cognitive and affective skills would alter intrinsic functional and structural organizational axes in a systematic and domain-specific manner. We evaluated longitudinal changes in MRI-derived cortical functional gradients and qT1 profiles as well as their interrelationship in the context of the 9-month ReSource study (25). We demonstrated intrinsic functional and microstructural plasticity that varied as a function of distinct social mental trainings, and that can predict training-related behavioral change. In particular, training attention/mindfulness, emotion/motivation, and socio-cognitive skills led to differential changes in a-priori network integration/segregation anchored in the secondary gradient differentiating sensory modalities. Moreover, functional changes showed correspondence to microstructural changes as a function of cortical depth and content of training. In sum, here we provide longitudinal evidence of a relationship between human social behaviors and intrinsic cortical function and depth-varying microstructure.

In sum, combining a longitudinal mental training study with multi-modal imaging, we could show that mental TMs focusing on attention, socio-emotional and socio-cognitive skills resulted in differentiable change in intrinsic functional and microstructural organization. In line with prior work revealing differential changes in grey matter morphology after each of the three ReSource TMs in the same sample (26), the current work differentiates processes related to our ability of understanding the thoughts and feelings of ourselves and others within the intrinsic functional and microstructural organization of the human brain, as such, our findings are compatible with previous work suggesting a link between anatomical and functional hierarchies and global workspace theories of cognition (1-3). Although our work focused on healthy adults ranging from 20 to 55 years of age, our findings overall support the possibility that targeted mental training can enhance social skills and lead to co-occurring reconfigurations of cortical function and microstructure, providing evidence for experience-dependent plasticity.”

Moreover, to increase clarity in our Results section, we have now included the cortex-wide analyses for microstructure, before focusing on the specific networks and functional regions of interest.

“Overall training effects in microstructure as a function of cortical depth (Figure 3).

Having established alterations in integration and segregation of a-priori networks, we wished to evaluate the neurobiological relevance of these alterations. To do so, we investigated changes in cortical microstructure as a function of cortical depth, motivated by the idea that intrinsic functional changes may be anchored in microstructural plasticity and that these changes may occur in a depth-varying manner (4). Overall, ReSource training led to decreased qT1 values, i.e. increased myelination, in both TC1 and TC2 relative to RCC over the nine months training time, in all a-priori functional networks in particular in deeper microstructural compartments, whereas RCC showed subtle increases of qT1, suggesting decreased myelination (Supplementary Table 15, Supplementary Figure 5). Studying training-specific effects, we observed marked changes in cortical microstructure following 3-months-long mental training across domains (all FDRq<0.05). Presence showed marked increases in qT1 in posterior areas in superficial depth compartments, and marked decreases in qT1 in prefrontal and occipital regions that showed increased spatial extent as a function of cortical depth. Conversely, Affect resulted in extended decreases in qT1 in mid and deep depth compartments, in particular in bilateral frontal areas extending to parietal lobe, bilateral posterior cingulate, left fusiform gyrus and right insula. Perspective showed largely decreases in qT1 in superficial depths in parietal-temporal, precuneus, and sensory-motor areas, and an increase in qT1 in left prefrontal regions. Patterns were similar when comparing the TMs against each other, highlighting the differentiation between superficial and deep depth-varying changes between Perspective and Affect and medial prefrontal qT1 decrease following Presence relative to Perspective and Affect as well as Affect TC3 and RCC (in particular in case of TC1, Figure 3 —figure supplement 2).”

2. The data are analyzed and presented on multiple levels of granularity, which is very impressive but in the meantime makes the Results section hard to follow. For example, the findings for longitudinal connectivity changes are discussed in terms of (1) eccentricity of the whole brain, (2) each of the three gradients, (3) a priori functional networks, and combinations of the above (e.g., changes along gradient 3 of the a priori attention network). I would recommend the authors trim any redundant analyses or clarify the distinction between these analytical levels and the significance of including all.

We indeed investigated changes in brain functional and structural organization using a combination of cortex-wide approaches and targeted functional networks that mirror the functions trained in the respective *ReSource* training modules. We opted for a multi-level approach, initially capturing changes in functional organization using the first three cortex-wide gradients, followed by a specific analysis of potential functional networks involved. We ran a similar analysis for microstructure *i.e.,* first cortex-wide, and then within *a-priori* networks. For behavioral prediction, we combined all metrics summarized in the *a-priori* networks. We believe this multi-level approach provides a comprehensive and informative overview of brain reorganization following mental training, with each level of granularity providing complementary insights that would be overlooked or result in unbalanced interpretation when focusing on one granularity level alone. We have further clarified the different analytical steps at the end of the introduction and at the beginning of each result paragraph.

Introduction:

“Here, we leverage the ReSource study dataset to assess whether the targeted training of attention-interoception, socio-affective, and socio-cognitive skills can lead to domain-specific reorganization of (i) intrinsic function (as indexed by resting-state fMRI connectivity gradient analysis), and (ii) cortical microstructure (as indexed by quantitative T1 relaxometry, probed along the direction of cortical columns (20-22)). Such results would be in line with prior observations suggesting coupled change in brain structure and function (23, 24), and would help to gain insights into the association between social skills and models of brain organization. Longitudinal analyses of subjects-specific measures of functional integration and segregation evaluated whether these changes corresponded to corresponding change in intracortical microstructure. We also tested for associations to behavioral change in attention, compassion, and ToM markers using machine learning with cross-validation, to evaluate behavioral relevance at the individual level.”

Results:

“Embedding of socio-affective and -cognitive functions in cortical brain organization (Figure 1).

Our work examined changes in brain function and microstructure following social and cognitive mental training. We analyzed resting-state functional MRI (fMRI) measures, myelin-sensitive quantitative T1 (qT1) relaxometry, and behavioral data from 332 adults studied in the ReSource Project (25). The preregistered trial (https://clinicaltrials.gov/ct2/show/NCT01833104) involved three 3-month long TMs: (i) Presence, targeting interoception and attention, (ii) Affect, targeting empathy and emotion, and (iii) Perspective, targeting ToM. To gain a system-level understanding of brain changes associated with each TM, we took a multi-level approach, combining cortex-wide exploratory analyses of changes in functional and microstructural organization, with an investigation of a-priori defined functional networks hypothesized to be targeted by each TM, behavioral prediction of behaviors implicated in each domain.”

“Mental training-specific change in functional organization (Figure 2).

We first tracked training-related longitudinal changes in functional organization using a holistic and cortex-wide approach through probing the combination of functional gradients 1-3 in functional eccentricity following the different ReSource TMs. Following, we investigated specificity of effects in terms of functional gradient and a-priori functional networks associated with the TMs.”

“Overall training effects in microstructure as a function of cortical depth (Figure 3).

Having established alterations in integration and segregation of a-priori networks, we evaluated the neurobiological relevance of these alterations. We investigated changes in cortical microstructure as a function of cortical depth, motivated by the idea that intrinsic functional changes may be anchored in microstructural plasticity that occurs in a depth-varying manner (4).”

“Corresponding changes in functional organization and intra-cortical microstructure (Figure 4).

Having shown alterations in functional and microstructural organization following social mental training, we evaluated corresponding changes in cortical microstructure. A multilevel approach was chosen. First, we evaluated whether the regions observed in functional reorganization in Presence versus Perspective would also show microstructural change. Second, we studied training-specific microstructural alterations in a-priori functional networks. Third, we evaluated the spatial correlation between functional and structural organization as a function of cortical depth.”

“Functional eccentricity and intracortical microstructure predict social cognitive performance (Figure 5).

Last, we evaluated whether alterations in cortical microstructure and function following mental training could predict behavioral changes in domains targeted by the TMs. To model changes in brain functional and structural organization, we focused on functional eccentricity, the three gradients, and microstructural depth divided in upper, mid, and deep compartments averaged within a-priori functional networks.”

3. Following the previous comment, it would be helpful to briefly introduce the importance of extracting and examining three gradients, especially G3, which in my experience is less seen in the literature.

We are happy to further motivate the extraction of the first three gradients. Theoretically, G1-G3 have been well established in terms of their potential cognitive relevance (11, 27-29). Whereas the G1 is associated with a differentiation of perception/action processes from abstract cognition, G2 differentiates visual from sensory/motor cortices. G3 differentiates control, multiple demand from default mode networks, and is linked to on- and off-task functional processes (28). These three gradients explain >50% of variance of the functional connectome. We now further clarified the approach in the result section.

“To investigate changes in intrinsic functional organization following different types of social and cognitive mental training we focused on changes within a 3D framework of functional axes, explaining in total more than 50% of variance within the functional connectome and well established in terms of potential cognitive relevance (11, 27-33). These axes differentiate primary from transmodal cortices (sensory/motor versus abstract cognition, principle gradient, G1), and within this axis further differentiation of visual from sensory-motor regions (secondary gradient, G2), and multiple demand and from default networks (tertiary gradient, G3). To synoptically assess changes within this functional framework we combined the first three gradients into a marker of functional eccentricity, similar to previous work (34). Here regions at either end of the gradient have a high eccentricity, a value based on the average of the three gradients. Following, we investigated gradient-specific effects.”

4. Please provide more details for statistical analysis in the methods section. For example, in the comparison of presence vs. perspective, is it overall changes after presence training vs. after perspective training, or changes in the selective group that had perspective training after presence?

To compare the effects of the different TMs, we combined all groups that underwent the particular 3-months-long TM using linear mixed effect models with subject as random effect. Thus, in the case of *Presence* versus *Perspective*, we included both TC1 and TC2 that completed *Presence* between baseline and the first measurement point, and for *Perspective* either between the first and second measurement point (TC2) or the second and third measurement point (TC1). Such a design would, at least in part, account for sequence effects. We now further clarified this in the *Methods* and *Results*, by annotating the TMs with the timepoints and training cohorts considered per main analyses.

Methods:

“Main analysis contrasts the TMs in TC1 and TC2 against each other to account for possible training general effects whilst controlling for subject as random effect in the linear model, and age and sex. In follow-up analyses we compared TMs against retest control cohorts, active control cohort (TC3), and for training cohorts independently.”

M = 1 + A + S + TM + random(Subject) + I

Results:

“In our main analyses, we compared TMs against each other focusing on TMs completed by TC1 and TC2, i.e. Presence (T_0_→T_1_, TC1 and TC2), Affect (T_1_→T_2_, TC1 and T_2_→T_3_, TC2), Perspective (T_2_→T_3_, TC1 and T_1_→T_2_, TC2) and supplementary investigations including also TC3 that only completed a socio-affective training and retest control cohorts.”

5. 12 cortical layers were sampled from the qT1 map acquired at 1mm isotropic resolution. Given that many changes were observed near the pial surface, such as Figure 3A-ii. Please discuss if and how the effects of partial volume averaging were accounted for. Were all qT1-related analyses controlled for cortical thickness?

Indeed, partial volume effects could in theory lead to changes in superior compartments of the qT1 signal. However, given the potential shared effects of microstructure and cortical thickness we did not include the regression of cortical thickness in our main analyses. Yet, when controlling for cortical thickness, we found similar effects (Supplementary Table 37-39). Moreover, when comparing spatial similarity of changes in intrinsic function and microstructure as a function of cortical depth, we found that in particular changes in the upper compartment corresponded to changes in functional eccentricity and G2 and G3 following *Presence*. Notwithstanding this could also be a correlation of noise, and further work is needed using ultra-high-resolution paradigms to understand the relationship between depth-varying microstructure and function. We added these considerations to the *Discussion*.

“Our work highlighted depth-dependent changes in cortical microstructure, with associations to intrinsic functional and behavioral change that go above and beyond cortical thickness alterations. Yet, we cannot exclude that part of the effects also include partial volume averaging, given the resolution of the data and study set-up. Further work will benefit from including longitudinal paradigms with sub-millimeter microstructural and intrinsic functional markers to further understand the interplay between depth-varying microstructure, its plasticity, and intrinsic brain function.!

6. I must admit that I'm a bit confused about the qT1 layer results, especially those in Figure 3B. The colored lines seem to indicate across-depth qT1 changes in each functional region. Then why would there be four lines for all networks? For example, why is there a green line, indicating perspective network, in the plot for attention network results? More details in the figure caption would be helpful.

To clarify, we depicted the average effects in each training module, TM (color code: *Presence*: yellow; *Affect*: red; *Perspective*: green) and retest control (blue) as a function of cortical depth as ‘spaghetti plots’. The boxes on the right of each spaghetti plot display the statistics (t-values) of the difference between TMs, with the contrast color coded as upper minus lower TM (defined by color). We have now added this explanation to the legend to further clarify and added further clarifying annotations in the figure.

“Figure 4. Dissociable microstructural alterations following mental training. A). TM specific changes in cortical microstructure; i. probing depth-specific cortical microstructure; ii. qT1 in regions showing eccentricity change (y-axis: depth, x-axis: qT1 change); B) Network-specific change in cortical microstructure as a function of depth, mean change per TM, p_FDR_<0.05 have black outline (y-axis: depth, x-axis: qT1 change). The boxes on the right of each plot display the statistics (t-values) of the respective difference between TM, with the contrast color coded as upper minus lower TM (defined by color); **C**) Correspondence of functional versus microstructural change; i. Spatial correlation of mean alterations in each TM, black outline indicates p_spin_<0.05, as a function of cortical depth.”

Reviewer #2 (Recommendations for the authors):In the manuscript entitled "Functional and microstructural plasticity following social and interoceptive mental training" by Valk et al., the authors present an analysis of the ReSource study data wherein they examine how gradients of functional connectivity and microstructure change following mindfulness and cognitive training. Their findings indicate that individuals' functional connectivity and cortical microstructure change longitudinally in response to these interventions, providing compelling evidence that interventions such as mindfulness change the brain's structure. Additionally, the authors used these changes to predict measures of attention, compassion, and perspective taking.I was invited to review this manuscript after an initial round of revisions. However, I was not provided a manuscript with changes tracked, which meant I could not review the paper before and after the authors' changes. Additionally, the authors provided only a summary of their changes to the original reviewers, rather than a point-by-point response. This made it unclear which of the original reviewers' comments were addressed and which were not. As such, instead of offering new comments, I have chosen to go through the original reviewers' comments and pick out ones that I (i) think are important and (ii) do not believe have been addressed by the authors.

We thank the Reviewer for noting this and apologies for this omission in the re-submission of the work. As we resubmitted the paper that was significantly altered beyond the requests of the Reviewers, including another modality and a different and updated machine learning pipeline, we decided a point-by–point response would not make sense in the current case.

"The word 'module' is used to describe the different behavioral training regimens that participants completed. As the authors are likely aware, this word is also commonly used to describe network structure. This makes phrases such as "module-specific behavioral change" and "behavioral changes across modules" difficult to parse. The authors may want to consider revising their use of the term module."I agree with R1 here. The use of the term 'module' conflicts with the broader fields typical use of the term (referring to modules within graph topology). Could the authors consider rephrasing to 'training module' or TM?

We thank the Reviewer for the suggestion. We have replaced the term Module wherever possible with ‘Training-module (TM)’.

"The central goal of the study is unclear. If the objective was to, as stated in some places, determine changes in integration/segregation of networks, the approach seems too indirect. A direct approach would evaluate these properties with graph theory. The authors do provide some results in that direction, but it is not clear why the results are secondary and mainly used to lend support to their gradients approach."I agree with R2 here. In general, I found the link between the authors measure of eccentricity and the twin pillars of functional integration and segregation to be unclear. The authors state:"we calculated region-wise distances to the center of a coordinate system formed by the first three gradients G1, G2, and G3 for each individual [based on the Schaefer 400 parcellation (67)]. Such a gradient eccentricity measures captures intrinsic functional integration (low eccentricity) vs segregation (high eccentricity) in a single scalar value (68)."I understand that this statement includes a relevant citation, but I think there is room for more intuition building here. What does high eccentricity correspond to in this distance calculation? Is it high distance from the center of a coordinate system? If so, why does high distance from center correspond to a segregated brain? Why does low distance from center correspond to an integrated brain? Like R2, I found it difficult to reconcile this gradient-distance based metric with my graph topology understanding of segregation and integration.

We are happy to further clarify the study goal. We aimed to understand how social and cognitive mental training impacts intrinsic functional and microstructural brain organization. Various lines of evidence have shown brain structure and function scaffold social and cognitive skills, in regions placed in unique positions along a cortical hierarchy differentiating perceptual-sensory from more abstract cognitive functions(11, 13, 35), but the inverse, how social and cognitive skills shape brain structure and function is not yet understood. To assess changes in functional organization we used a gradient-based framework, focusing on the first three gradients. These three gradients explain over half of the variance in the functional connectome and have been cognitively interpreted in previous work. Together they illustrate a differentiation between perceptual/action processes from more abstract cognitive functions, though each with axis-specific nuances. For example, the first gradient differentiates sensory/motor from DMN regions, whereas the second gradient typically differentiates visual from sensory-motor networks. Conversely the third gradient differentiates multiple demand networks from default networks. To gain a broad insight into changes of these gradients combined, we created an eccentricity metric. Here regions that are at either extreme of one of the three gradients – e.g. show most differentiated connectivity profiles along the respective hierarchies get a high eccentricity value, whereas those that are in the middle of the gradient have a value close to zero (the eccentricity is based on the square root of the absolute gradient loading value). As such this combined metric gives a first estimate of connectome organization and its changes following training. Moreover, the gradient framework allowed us to further investigate whether changes occurred in a particular organizational axis. We have now updated the rationale and motivation behind the eccentricity value in the Results section.

Results:

“To investigate changes in intrinsic functional organization following different types of social and cognitive mental training we focused on changes within a 3D framework of functional axes, explaining in total more than 50% of variance within the functional connectome and well established in terms of potential cognitive relevance (11, 27-33). These axes differentiate primary from transmodal cortices (sensory/motor versus abstract cognition, principle gradient, G1), and within this axis further differentiation of visual from sensory-motor regions (secondary gradient, G2), and multiple demand and from default networks (tertiary gradient, G3). To synoptically assess changes within this functional framework we combined the first three gradients into a marker of functional eccentricity, similar to previous work (34). Here regions at either end of the gradient have a high eccentricity, a value based on the average of the three gradients. Following, we investigated gradient-specific effects.”

Prediction analyses:R1 stated:"Although potentially quite interesting, it is not clear that the connectivity-based prediction of behavioral changes is very robust. Effect sizes are small to medium, the methods used for these analyses are prone to data leakage (and steps to protect against these problems are not described), effect size estimates are based on cross-validation alone (as opposed to out-of-sample tests), and there are many experimenter degrees of freedom."R3 stated:"In the prediction work it's not clear what exact behavioral measures were tested. There is a description on page 19 under behavioral markers starting on line 33, but it would be helpful if this was described earlier (maybe this is a journal formatting problem). However, in this section there is a description of how the measure for compassion and ToM were obtained but no description of the attention score."And"It wasn't clear to me if the specific networks tested did better at predicting the related tasks or if cross-network-task predictions were investigated. For example, you measured the extent to whether G1-G3 in the attention network predicted attention scores. Did these same G1-G3 values predict socio-affective or ToM scores? Did the socio-affective network predict socio- cognitive skills better than attention and ToM? In other words what happened if you mixed and matched these across networks – how critical are the specific network definitions to the prediction?"Presently, I do not think these comments about the authors' prediction analyses have been addressed. Moreover, I too have concerns about the authors' approach. For example, the authors state "Before running our model, we regressed out age and sex from the brain markers." Here, it's unclear whether this nuisance regression was done in a leakage resistant way or not. This sentence implies that age and sex were regressed out of all the data in a single step prior to running the prediction model. If this is the case, this approach will cause leakage and spuriously boost prediction performance. To avoid this, the authors should consider incorporating nuisance regression into their cross-validation model, wherein nuisance models are fit to the training data and applied to the test data.

We thank the Reviewer for this remark. In the current framework we have included nuisance regression separately for the training and test sample, and we apologize for the unclear description. We now updated the description of the analyses.

Results.

“We incorporated age and sex regression into the cross-validation model, to avoid leakage.”

Moreover, to test the previous suggestions of the Reviewers, we now ran the prediction model on randomized test data, as well as compassion and ToM change scores within the respective TM and participant set. We found that the domain-specific prediction models, e.g. attention in Presence, compassion in Affect, ToM in Perspective, albeit of modest strength, provided significantly better prediction than on random/ non-domain behavioral outcomes. We also included this into our results.

Results:

“To further test our predictive models, we evaluated model performance on random test data, as well as non-domain behavioral scores. We found that in all cases, the domain-specific TM model performed best on test data of the respective TM (p<0.001).”

Methods.

“We adopted a supervised framework with cross-validation to predict behavioral change based on change in functional and microstructural organization within five functional networks. We aimed at predicting attention, compassion, and perspective-taking (Figure 4). Before running our model, we regressed out age and sex from the brain markers within the cross-validation loop to avoid leakage. We utilized 5-fold cross-validation separating training and test data and repeated this procedure 100 times with different sets of training and test data to avoid bias for separating subjects. Following we performed an elastic net cross validation procedure with alphas varying from 0.0001-1, ratio 1.0, making it a lasso regression. We used sequential feature selection to determine the top 20% of features based on mean absolute error without cross validation. Linear regression for predicting behavioral scores was constructed using the selected features as independent variables within the training data (4/5 segments) and it was applied to the test data (1/5 segment) to predict their behavioral scores. The prediction accuracy was assessed by calculating Pearson’s correlation between the actual and predicted behavioral scores as well as their negative mean absolute error, nMAE. To further assess specificity of the behavioral prediction models, we evaluated Pearson’s correlation between actual and predicted scores based on randomized scores, as well as using the model to predict out-of-TM data (e.g. for attention in Presence; compassion in Affect and ToM in Perspective; for compassion in Affect, attention in Presence and ToM in Perspective; for ToM in Perspective; attention in Presence and compassion in Affect).”

Reviewer #3 (Recommendations for the authors):The study by Valk and colleagues investigated the impact of various forms of social mental training on resting-state functional connectivity and myeloarchitecture using a large-scale longitudinal multimodal dataset. The examination of these changes in combination is particularly noteworthy.However, a few key points should be addressed to further improve the study.[Line 140] The use of task-based networks to summarize changes across the cortex may be problematic, as the metrics averaged within each network may be influenced by small clusters, rather than reflecting the entire network. The FDR-survived changes in Attention and Interoception networks may be due to overlap in vertices, rather than network properties. To avoid this, it may be beneficial to consider alternative methods for summarizing whole-cortex changes.

Thanks for this suggestion! Indeed, the networks overlap. As regions show different network affiliation structure during different cognitive tasks, we believe their overlap should also be perceived in this perspective and removing overlap would also significantly alter interpretation of the networks. Conceptually, it is hard to differentiate the role of interoception and attention within the current framework, as the Presence training module targets both. Hence, we would refrain from changing our approach in the main analyses, also given we provide whole-brain assessments in parallel to *a-priori* network-based approaches.

However, to directly address the Reviewers concern, we additionally created non-overlapping networks to see if regions specific to the respective networks show differential change in functional and microstructure. Findings were highly comparable.

**Author response table 1. sa2table1:** Eccentricity and non-overlapping networks (Figure 2).

	Presence-Perspective	Presence-Affect	Perspective-Affect			
attention	2,776	p=0,006	1,451	p=0,147	-1,624	p=0,105
interoception	2,637	p=0,009	1,279	p=0,201	-1,641	p=0,101
emotion	0,034	p=0,973	-0,353	p=0,724	-0,377	p=0,706
empathy	1,700	p=0,090	0,510	p=0,610	-1,360	p=0,174
ToM	1,901	p=0,058	1,172	p=0,242	-0,943	p=0,346

**Author response table 2. sa2table2:** Microstructure and non-overlapping networks as a function of cortical depth in qT1 (Figure 4).

Presence-Perspective									
attention		interoception	emotion		empathy		ToM		
t-value	p-value	t-value p-value	t-value	p-value	t-value	p-value	t-value	p-value	
5,680	0,000	3,673	0,000	-0,763	0,446	3,737	0,000	4,995	0,000
5,593	0,000	4,454	0,000	-0,824	0,410	3,892	0,000	4,753	0,000
5,185	0,000	4,743	0,000	-0,985	0,325	3,826	0,000	4,086	0,000
4,565	0,000	4,582	0,000	-1,211	0,226	3,601	0,000	3,171	0,002
3,845	0,000	4,135	0,000	-1,444	0,149	3,306	0,001	2,331	0,020
3,119	0,002	3,509	0,000	-1,605	0,109	3,020	0,003	1,741	0,082
2,436	0,015	2,770	0,006	-1,656	0,098	2,690	0,007	1,440	0,150
1,852	0,065	1,920	0,055	-1,658	0,098	2,320	0,021	1,321	0,187
1,434	0,152	1,038	0,300	-1,696	0,090	1,952	0,051	1,268	0,205
1,215	0,225	0,224	0,823	-1,796	0,073	1,683	0,093	1,245	0,214
1,161	0,246	-0,446	0,656	-1,899	0,058	1,560	0,119	1,234	0,218
1,216	0,225	-0,949	0,343	-2,018	0,044	1,546	0,123	1,214	0,225
									
Presence-Affect									
3,868	0,000	2,685	0,007	0,913	0,362	3,055	0,002	3,133	0,002
4,103	0,000	3,179	0,002	0,960	0,338	3,257	0,001	3,036	0,003
4,091	0,000	3,420	0,001	0,915	0,360	3,361	0,001	2,820	0,005
3,911	0,000	3,447	0,001	0,828	0,408	3,403	0,001	2,560	0,011
3,656	0,000	3,313	0,001	0,809	0,419	3,432	0,001	2,395	0,017
3,386	0,001	3,090	0,002	0,892	0,373	3,506	0,000	2,372	0,018
3,118	0,002	2,847	0,005	1,041	0,298	3,588	0,000	2,418	0,016
2,867	0,004	2,618	0,009	1,201	0,230	3,552	0,000	2,434	0,015
2,663	0,008	2,404	0,017	1,268	0,205	3,363	0,001	2,381	0,018
2,552	0,011	2,181	0,030	1,251	0,211	3,131	0,002	2,295	0,022
2,526	0,012	2,017	0,044	1,169	0,243	2,957	0,003	2,212	0,027
2,555	0,011	1,887	0,060	0,992	0,322	2,876	0,004	2,176	0,030
									
Perspective-Affect									
-2,447	0,015	-1,406	0,160	1,709	0,088	-1,124	0,262	-2,418	0,016
-2,127	0,034	-1,778	0,076	1,821	0,069	-1,101	0,271	-2,249	0,025
-1,697	0,090	-1,858	0,064	1,953	0,051	-0,932	0,352	-1,733	0,084
-1,196	0,232	-1,658	0,098	2,115	0,035	-0,648	0,517	-0,989	0,323
-0,660	0,509	-1,301	0,194	2,350	0,019	-0,299	0,765	-0,233	0,816
-0,132	0,895	-0,837	0,403	2,606	0,009	0,084	0,933	0,386	0,699
0,352	0,725	-0,270	0,787	2,805	0,005	0,524	0,600	0,759	0,448
0,744	0,457	0,432	0,666	2,957	0,003	0,894	0,372	0,904	0,367
1,002	0,317	1,182	0,238	3,062	0,002	1,111	0,267	0,910	0,363
1,133	0,258	1,854	0,064	3,153	0,002	1,182	0,238	0,852	0,394
1,166	0,244	2,426	0,016	3,186	0,002	1,148	0,252	0,784	0,433
1,134	0,257	2,849	0,005	3,146	0,002	1,085	0,278	0,772	0,441

[Line 217] The motivation for the selective analysis is clear, but the overall effects of the training were unclear until later in the manuscript (Line 234 and Supplementary Figure 6). It may be beneficial to describe the overall effects of training upfront.

Thanks for the suggestion! We agree with the Reviewer and moved the overall effects of the training to the main Results section.

“Overall training effects in microstructure as a function of cortical depth (Figure 3).

Having established alterations in integration and segregation of a-priori networks, we evaluated the neurobiological relevance of these alterations. We investigated changes in cortical microstructure as a function of cortical depth, motivated by the idea that intrinsic functional changes may be anchored in microstructural plasticity that occurs in a depth-varying manner (4). Overall, ReSource training led to decreased qT1 values, i.e. increased myelination, in both TC1 and TC2 relative to RCC over the nine months training time, in all a-priori functional networks in particular in deeper layer microstructure, whereas RCC showed subtle increases of qT1, suggesting decreased myelination (Supplementary Table 15, Supplementary Figure 5). Studying training-specific effects, we observed marked changes in cortical microstructure following 3-months-long mental training across domains (all FDRq<0.05). Presence showed marked increases in qT1 in posterior areas in superficial depth compartments, and marked decreases in qT1 in prefrontal and occipital regions that showed increased spatial extent as a function of cortical depth. Conversely, Affect resulted in extended decreases in qT1 in mid and deep depth compartments, in particular in bilateral frontal areas extending to parietal lobe, bilateral posterior cingulate, left fusiform gyrus and right insula. Perspective showed largely decreases in qT1 in superficial depths in parietal-temporal, precuneus, and sensory-motor areas, and an increase in qT1 in left prefrontal regions. Patterns were similar when comparing the TMs against each other, highlighting the differentiation between superficial and deep depth-varying changes between Perspective and Affect and medial prefrontal qT1 decrease following Presence relative to Perspective and Affect as well as Affect TC3 and RCC (in particular in case of TC1, Figure 3 —figure supplement 2).”

[Line 268] The predictive performance of the models was not explicitly tested to determine if it was above the chance level. For a nonparametric test, it may be useful to calculate the null distribution of chance level performance through the use of permuted pairs of predictors and targets.

We thank the Reviewer for the suggestion and have now run the models using a permutation approach for behavioral scores.